# The DEAD-Box RNA Helicase Ded1 Is Associated with Translating Ribosomes

**DOI:** 10.3390/genes14081566

**Published:** 2023-07-31

**Authors:** Hilal Yeter-Alat, Naïma Belgareh-Touzé, Emmeline Huvelle, Josette Banroques, N. Kyle Tanner

**Affiliations:** 1Expression Génétique Microbienne, Université de Paris Cité & CNRS, IBPC, 13 Rue Pierre et Marie Curie, 75005 Paris, France; hilal.yeter@ibpc.fr (H.Y.-A.); emmeline.huvelle@i2bc.paris-saclay.fr (E.H.); josette.banroques@ibpc.fr (J.B.); 2Institut de Biologie Physico-Chimique, Paris Sciences et Lettres University, CNRS UMR8261, EGM, 75005 Paris, France; 3Laboratoire de Biologie Moléculaire et Cellulaire des Eucaryotes, UMR8226 CNRS, Institut de Biologie Physico-Chimique, Sorbonne Université, 13 Rue Pierre et Marie Curie, 75005 Paris, France; naima.belgareh@ibpc.fr

**Keywords:** Ded1/DDX3, UV-A, ATPase, DEAD-box, qtPAR-CLIP

## Abstract

DEAD-box RNA helicases are ATP-dependent RNA binding proteins and RNA-dependent ATPases that possess weak, nonprocessive unwinding activity in vitro, but they can form long-lived complexes on RNAs when the ATPase activity is inhibited. Ded1 is a yeast DEAD-box protein, the functional ortholog of mammalian DDX3, that is considered important for the scanning efficiency of the 48S pre-initiation complex ribosomes to the AUG start codon. We used a modified PAR-CLIP technique, which we call quicktime PAR-CLIP (qtPAR-CLIP), to crosslink Ded1 to 4-thiouridine-incorporated RNAs in vivo using UV light centered at 365 nm. The irradiation conditions are largely benign to the yeast cells and to Ded1, and we are able to obtain a high efficiency of crosslinking under physiological conditions. We find that Ded1 forms crosslinks on the open reading frames of many different mRNAs, but it forms the most extensive interactions on relatively few mRNAs, and particularly on mRNAs encoding certain ribosomal proteins and translation factors. Under glucose-depletion conditions, the crosslinking pattern shifts to mRNAs encoding metabolic and stress-related proteins, which reflects the altered translation. These data are consistent with Ded1 functioning in the regulation of translation elongation, perhaps by pausing or stabilizing the ribosomes through its ATP-dependent binding.

## 1. Introduction

DEAD-box proteins are ATP-dependent RNA binding proteins and RNA-dependent ATPases that are capable of remodeling RNP complexes, acting as RNA chaperones to alter RNA structures and as helicases to unwind short RNA duplexes, but they are nonprocessive and incapable of unwinding extensive regions of base pairing (reviewed by [1,2,3,4,5]). DEAD-box proteins have a highly conserved core structure consisting of two domains with structural homology to recombinant protein A (RecA-like) and highly variable amino- and carboxyl terminal sequences. In the absence of ATP, the two RecA-like domains are unconstrained (“open” complex), with low affinity for the RNA. However, with bound ATP, the two domains form a constrained “closed” complex with high affinity for the RNA. In the solved crystal structures with bound ligands, RecA-like domain 2 binds the 5′ end of the RNA in the form of an A helix, which can be either single-stranded or as part of a duplex (reviewed by [6,7]). In contrast, the RNA binding motifs Ib and GG of RecA-like domain 1, in the presence of bound ATP, disrupt the A form helix due to steric hindrance to form a kink in the phosphodiester backbone. This is considered the mechanism for duplex unwinding by the proteins, but the kinking also effectively locks the proteins on the RNA and prevents sliding. Indeed, the DEAD-box protein Ded1 bound to the nonhydrolyzable ATP analog ADP-BeF_x_ forms long-lived complexes on RNA in vitro [8]. Thus, DEAD-box proteins can function as ATP-dependent RNA clamps for the assembly of RNPs and to provide directionality to multistep processes, such as in ribosomal biogenesis or splicing [3,9].

The yeast *Saccharomyces cerevisiae* expresses 25 different DEAD-box proteins, most of which are essential and non-interchangeable even when overexpressed [10,11]. Humans have 37 different proteins, of which many are clear homologs of the yeast proteins [12,13]. One such homolog is the Ded1/DDX3 protein (reviewed by [14,15,16,17,18]). Indeed, the mammalian DDX3X is capable of complementing a yeast strain deleted for the essential *DED1* gene ([19] and references therein). Thus, the yeast and mammalian proteins are functional orthologs, where the biological roles have been conserved over evolution, and it means that work on the different proteins is comparable.

Ded1 was first found to be an extragenic suppressor of a *prp8* mutation associated with splicing [20]. It was later implicated in the transcription of polymerase III RNAs [21], as a general translation initiation factor [22,23], in 48S pre-initiation complex (PIC) ribosome scanning [24], in yeast L–A virus replication [25], and in the response to TORC1 signaling [26]; moreover, it is found in cytoplasmic foci (processing bodies or stress granules; P-bodies and SG, respectively) containing translation-inactive mRNAs in association with the DEAD-box protein Dhh1, which is involved in mRNA degradation [19,27,28,29,30,31]. Similarly, DDX3 is an important factor in the replication of HIV-1 and other viruses (reviewed by [16,32]), in the formation of cytoplasmic foci [33], and in translation [34,35,36]. The Ded1/DDX3 subfamily of proteins are implicated in developmental and cell cycle regulation (reviewed by [14,15,16,17,18,33,37]). We previously showed that Ded1 is an mRNP CAP-associated factor that interacts with proteins of both the nuclear cap binding complex (CBC) and the cytoplasmic eIF4F complex, and that these interactions involve protein–protein interactions that are independent of the RNA; nevertheless, many of these factors stimulate the RNA-dependent ATPase activity of Ded1 [19]. Moreover, Ded1 is exported from the nucleus through the Xpo1/Crm1 nuclear pore complex [19,38]. Similar results were obtained with mammalian DDX3 [33,39]. These results suggest that Ded1 may interact with mRNAs in a precocious fashion, perhaps even during transcription.

Currently, the Ded1/DDX3 proteins are considered translation initiation factors that play a role in the scanning of the 48S PIC to the AUG start codon and subsequent formation of the 48S initiation complex (IC) by removing RNA secondary structures that might impede the scanning ribosomes [40,41,42,43,44]. This proposed role is considered particularly important for mRNAs with long, highly structured 5′ untranslated regions (UTRs). Thus, DDX3 is needed for expressing mRNAs with complex 5′ UTRs [34]. A similar role was proposed for the DEAD-box initiation factor eIF4A, but recent cryo-electron microscopy work indicates that the eIF4F complex containing eIF4A is located 5′ to the scanning ribosomes [45]. In this case, the ATP-dependent clamping mechanism of eIF4A could provide directionality to the scanning ribosomes by preventing backward sliding, as originally proposed by Spirin for the Brownian ratchet model of ribosome scanning [46,47]. This role would be particularly important for mRNAs with long 5′ UTRs as random Brownian motion would impose much longer scanning times for the 48S PICs before the AUG start codon is found. This does not rule out additional helicase activity of eIF4A as the eIF4F complex could unwind downstream structures by pulling on the mRNAs or by interacting with downstream sequences when the mRNA is looped back through the poly(A) interactions of PABP [45]. Yeast Ded1 could serve a similar role, but it has a much higher ATP-dependent affinity for RNA, and it is much more effective as an RNA helicase than eIF4A [11]. It is currently unknown where Ded1/DDX3 is located relative to the 48S PICs.

Recent work has used CLIP (crosslinking and immunoprecipitation), CRAC (crosslinking analysis of cDNAs), and ribosome profiling experiments to elucidate the RNA substrates and binding sites of Ded1 in vivo [40,41,44,48]. However, ribosome scanning can reveal indirect effects of the protein; moreover, we find that the short, 254 nm, ultraviolet (UV) light used in classical crosslinking experiments rapidly degrades Ded1. Finally, short UV light generates diffusible hydroxyl radicals and other reactive species in biological solutions that yield conditions that are not physiological [49]. Indeed, we find that cells are killed within seconds with 254 nm UV under our conditions. In this work, we used a modified photoactivable ribonucleoside CLIP (PAR-CLIP) technique, which we call quicktime PAR-CLIP (qtPAR-CLIP), that involves incorporating 4-thiouridine (4-thioU) into the yeast RNA and then irradiating with long UV light centered at 365 nm [50,51]. Notably, the 4-thioU can reveal nucleotide-specific resolution of the binding sites [51,52,53]. Moreover, yeast shows very few adverse effects from the qtPAR-CLIP conditions, which provide assurances that the crosslinking interactions were obtained under physiological conditions. We undertook crosslinking under both standard growth conditions and in the brief absence of sugar, which is known to quickly alter the profile of translated mRNAs. In contrast to previous published work, we find that Ded1 forms a majority of the crosslinks to the open reading frames (ORFs) of a subset of mRNAs that have highly variable 5′ UTRs, lengths of the ORFs, and expression levels. Many of these mRNAs encode factors involved in translation or in translation regulation. Moreover, many of the other crosslinked RNAs (e.g., tRNAs) are associated with translation regulation. Finally, we obtain significant crosslinking on the Gag and Gag–Pol RNAs of retrotransposons. These results indicate that Ded1 has a role as a translation elongation factor that might help to regulate the expression of certain mRNAs.

## 2. Materials and Methods

### 2.1. Constructs, Yeast Strains, and Manipulations

The wildtype BY4742 yeast strain (*Mat-α his3∆1 leu2∆0 ura3∆0 lys2∆0*; Euroscarf, Oberursel, Germany) was transformed with a plasmid expressing *DED1* gene under its own promoter and terminator in the YCplac111 *ars cen* plasmid containing the *LEU2* marker according to standard procedures [54]. This strategy enabled us to test various *DED1* mutants, including forms that were otherwise lethal. The expression levels of Ded1 were otherwise similar to that of the endogenous gene. The cells were grown in Synthetic Defined medium lacking leucine (SD-LEU), which we define as standard conditions (WT).

The *GFP* and *MCHERRY* plasmids were constructed by amplifying genes off the pYM27–*EGFP–KanMX4* and pFA6a–*mCherry–NatNT*2 plasmids, respectively. The PCR products were digested with XhoI and SalI, gel-purified, and cloned into the equivalent sites of the yeast plasmids p415, p416, and p413. The *DED1* and *ded1–DQAD* plasmids were as previously described [19]. These genes were subcloned into the SpeI and XhoI sites of *GFP*-p415, p414, and *MCHERRY*_p416. The *KAR2-RFP*_YIPlac204 was a gift from Benjamin Glick.

The *sec61-ts* and *sec62-ts* yeast strains were a generous gift from Ron Deshaies [55]. The *KAR2-RFP* strain was created by transforming the W303 (G49), *sec61,* and *sec62* strains with EcoRV-linearized *KAR2-RFP*_YIPlac204 containing the N-terminus of *KAR2* (135 bp) fused to *DsRedExpress2* with the *HDEL ER*-retention sequence (YIPlac204TKC-DsRed-Express2-HDEL; Addgene, Watertown, MA, USA).

Ded1 was purified as previously described [2]. The anti-Rabbit Ded1-IgG and pre-immune-IgG were produced by Covalab (Bron, France) using purified recombinant Ded1 for the former. The anti-mouse PGK1 IgG was obtained from Abcam (Cambridge, UK).

### 2.2. Growth Curves

The 4-thiouracil was purchased from Sigma-Aldrich (Darmstadt, Germany). Stock solutions of ~2 mg/mL were created by dissolving in deionized water and titrating to a pH of ~9.0 with 10 N NaOH. The solution was filter-sterilized and the final concentration was determined with a Uvikon 933 spectrophotometer using a molar extinction coefficient of 18,600 M^−1^cm^−1^ at 328 nm [56]. Overnight cultures were diluted to an OD_600_ of 0.15 mL^−1^cm^−1^ in SD-LEU medium and grown in duplicate for each concentration of 4-thiouracil in sterile 96-well microtiter plates (Falcon Tissue Culture 353072, Corning, NY, USA) with low evaporation lids in a Tecan (Männedorf, Switzerland) Infinite M200Pro microtiter reader at 30 °C with periodic mixing in the linear and orbital directions. Measurements were taken approximately every 10 min with a path length of 0.4–0.5 cm.

### 2.3. 4-thioU Incorporation and Crosslinking Conditions

The wildtype BY4742 yeast strain with the *DED1* gene under its own promoter and terminator in the YCplac111 plasmid was grown at 30 °C in 500 mL SD-LEU to an OD_600_ of ~0.4 mL^−1^cm^−1^ (exponential phase). The cultures were then made 300 μM in 4-thiouracil. As previously noted, the 4-thiouracil was more readily absorbed by the yeast than the nucleoside [57]. Cultures were incubated to an OD_600_ of 1.8–2.0 mL^−1^cm^−1^ (~5 h). We estimate that ~1.3% of the uridines were substituted for 4-thioU based on the ratio of absorption at 330 nm relative to 260 nm of the purified RNA and by using a molar extinction coefficient of 21,600 M^−1^cm^−1^ at 330 nm for 4-thioU [56]. Background absorption at 330 nm was determined using the purified RNA from cells not incubated with 4-thiouracil. Cells were harvested by centrifugation, quickly rinsed with cold water, and then resuspended in ~20 mL of ice-cold water to yield an OD_600_ equivalent to ~32 mL^−1^cm^−1^. In parallel experiments, cells were resuspended in medium lacking glucose (∆Glu) and grown for an additional 15 min at 30 °C prior to collection and resuspended in cold water. One third of the culture was set aside for transcriptome analyses and the other two thirds for the crosslinking.

For the qtPAR-CLIP experiment, cells incubated with 300 μM 4-thiouracil in SD-LEU medium were placed in 8.5 cm polystyrene Petri dishes, with covers, cooled on ice, and irradiated twice for 10 min with UV-A lamps centered at 365 nm in a UV Stratalinker 2400 (Stratagene, San Diego, CA, USA) with shaking between irradiations. The distance was 1–2 cm and flux was ~7 Joules/cm^2^ total. In contrast with other PAR-CLIP experiments, we used the polystyrene covers to block irradiation below 290 nm, which further reduced UV-induced protein degradation caused by the broad spectrum 365 nm lamps [53,58]. Cells (crosslinked and transcriptome) were then collected by centrifugation at 6000 rpm for 5 min at 4 °C in a Beckman (Villepinte, France) JA-12 rotor, washed with cold water, and then the cell pellets were frozen in liquid nitrogen and stored at −80 °C until needed.

### 2.4. Growth Viability and Protein Stability to UV Light

For growth viability experiments, the equivalent of 2 mL of cells with an OD_600_ of ~1 mL^−1^cm^−1^ were placed in 6 well culture plates (Cellstar, Greiner Bio-One, Les Ulis, France) cooled on ice and irradiated with 365 nm UV-A (with the polystyrene cover; ~7 J/cm^2^) or 254 nm UV-C (without cover; ~4.8 J/cm^2^) in a Stratalinker 2400 for the indicated times. One hundred µL of crosslinked cells were then placed into a 96-well, sterile, microtiter plates, serially diluted by a factor of 10, spotted on SD-LEU agar plates and incubated at 30°C for 3 days.

For the protein stability experiments, Ded1 was purified as previously described [2]. Aliquots of 1.6 µg of Ded1 in 20 µL 1X PBS buffer (Eurimedex, Souffelweyersheim, France), containing 1.06 mM KH_2_PO_4_, 3 mM Na_2_HPO_4_ and 154 mM NaCl, pH 6.7–7.0, were spotted on the rimed wells of the low-evaporation lids of 96 well tissue culture plates (Falcon) and irradiated for various times with 365 nm UV-A or 254 nm UV-C. Samples irradiated with UV-A were covered with polystyrene lids. Samples were recovered and the wells rinsed with 10 µL of PBS and combined with the samples.

Proteins from yeast cells were rapidly extracted as previously described with minor modification [59]. SDS sample buffer was added and the proteins were resolved by electrophoresis on a 10% SDS polyacrylamide gel (SDS-PAGE) and then stained with Instant Blue Coomassie stain (Expedeon-Abcam, Paris, France).

### 2.5. qtPAR-CLIP Protocol

The qtPAR-CLIP experiments were performed in triplicate based on the previously described PAR-CLIP method with some modifications to enhance the yield and reduce the number of steps [60]. The full protocol is as described here and a side-by-side comparison with the previous method is shown in Appendix A. The crosslinked cell pellets were resuspended in the equivalent of two cell-pellet volumes of lysis buffer containing 20 mM HEPES, pH 7.4, 150 mM NaCl, 5 mM MgCl_2_, 0.1 mM EDTA, 5 mM DTT, and protease inhibitor cocktail (Roche cOmplete-EDTA, Basel, Switzerland). One volume of 425–600 μm glass beads (Sigma-Aldrich) was added and the cells were lysed by vortexing in a FastPrep-24 (M.P. Biomedicals, Santa Ana, CA, USA) at 4 °C with a setting of 6.5 and using four cycles of 30 s on and 5 min rests on ice. Beads and cell debris were removed by centrifugation in an Eppendorf (Montesson, France) 5415R centrifuge at 4 °C at 6000 rpm, and then the lysates were further clarified by centrifugation at 13,200 rpm for 15 min at 4 °C. The protein concentration of the lysate was determined with the Bio-Rad (Hercules, CA, USA) Protein Assay using BSA as a standard.

We obtained much better Ded1 binding and recovery using Protein A Sepharose beads instead of magnetic beads. The Protein A Sepharose beads were prepared by first washing them twice with Ipp_300_ buffer containing 20 mM Tris-base, pH 7.4, 300 mM NaCl, 0.1% Triton-X100, and 1 mM MgCl_2_. Then, 50 μL of beads were incubated overnight at 4 °C in 10 volumes of Ipp_300_ buffer that was supplemented with 0.4 mg/mL BSA (Sigma-Aldrich), 0.4 mg/mL heparin (Sigma-Aldrich), and 20 μL of serum containing Ded1 or pre-immune immunoglobulin G (IgG). The beads were washed three times with Ipp_300_ buffer.

The equivalent of 300 μg of protein from the clarified lysates was added to each of 12 individual tubes containing ~50 μL of Protein A Sepharose beads bound with anti-Ded1 IgG or with the pre-immune serum control. The volumes were adjusted to 500 μL with Ipp_300_ buffer that was supplemented with 0.4 mg/mL BSA (Sigma-Aldrich), 0.4 mg/mL heparin (Sigma-Aldrich), and 1M urea to reduce nonspecific binding of proteins and RNA. We recovered very little material with the Protein A Sepharose beads using the pre-immune IgG.

The bead lysates were rotated over 2 h on a turning wheel at 4 °C. The bead immunoprecipitates were washed 2–3 times with cold Ipp_300_ buffer and then digested with 0.5 U/μL S1 nuclease (ThermoFisher Scientific #EN0321, Waltham, MA, USA) in 100 μL of the included S1 buffer containing 15% PEG8000 (Sigma-Aldrich) for 15 min at 30 °C in an Eppendorf Thermomixer Comfort with a cycle of 15 s mix at 1000 rpm and a 90 s rest. The nuclease digestions were stopped by adding one volume of Ipp_120_ buffer containing 20 mM Tris-base, pH 7.4, 120 mM NaCl, 5 mM EDTA, and 1% Triton-X100. The EDTA coordinated the Zn^2+^ ion needed by the S1 nuclease and quickly inactivated it. Beads were then washed once with Ipp_120_ buffer and twice with Ipp_300_ buffer. The adapter oligonucleotide was ligated at the 3′ end in a 100 μL reaction mix containing 10 units T4 RNA ligase (New England Biolabs, Évry-Courcouronnes, France), 1 pmole 3′ DNA adapter (Appendix A), manufacturer’s ligase buffer, 1 mM ATP, and 15% PEG8000 overnight at 16 °C in a Eppendorf Thermomixer Comfort with a cycle of 15 s at 1000 rpm and 90 s rests.

The ligation reactions were stopped by washing the Protein A Sepharose beads once with Ipp_120_ buffer and then twice with Ipp_150_ buffer (same as Ipp_300_ buffer except with 150 mM NaCl instead of 300 mM). The bound proteins with the crosslinked RNA were eluted twice with 300 μL of 0.1 M glycine, pH 2.3, for 15 min at 4 °C with mixing. The 12 fractions were combined and the acidity of the eluant was then adjusted to a pH of ~7 with NaOH. The proteins were eliminated by the addition of 1 mg/mL proteinase K (Sigma-Aldrich) supplemented with 1% Triton X-100, 0.5% SDS, and 5 mM CaCl_2_, and incubated at 55 °C for 35 min in an Eppendorf Thermomixer Comfort with a cycle of 15 s at 1000 rpm and a 90 s rest. The RNA was then recovered by making the solution 0.3 M in potassium acetate, extracting with an equal volume of water-saturated phenol, extracting twice with an equal volume of chloroform isoamyl alcohol (24:1), 50 μg/mL of glycogen carrier was added (Roche, Basel, Switzerland), and then the RNA was precipitated with 2.5 volume ethanol overnight at −20 °C.

RNAs were collected by centrifugation in an Eppendorf 5415R for 20 min at 4 °C at 13,000 rpm, followed by two washes with cold 70% ethanol. The resulting RNAs pellets were allowed to dry and then resuspended in 9 μL nuclease-free water and mixed with 21 μL RNA ligase reaction that contained 15 units T4 RNA ligase, 2 pmole DNA 5′ adapter (Appendix A), ligase buffer, 1 mM ATP, and 15% PEG8000. The ligation was performed for 2 h at 55 °C in an Eppendorf Thermomixer Comfort (15 s at 1000 rpm, 90 s rest). RNAs were purified with a NucleoSpin RNA Clean-up Kit (Macherey-Nagel, Düren, Nordrhein-Westfalen, Germany) and eluted with 35 μL water. We used 4.6 μL of ligated RNA to synthetize the cDNA library. The RNA RT primer (Appendix A) at 1 pmole was added and the samples were heated at 50 °C for 5 min. The reverse transcription reactions were completed with a Superscript III Reverse Transcriptase kit (Invitrogen, Waltham, MA, USA) according to the manufacturer’s instructions and incubated for 90 min at 50 °C. The RNAs were hydrolyzed by adding 40 μL of a solution containing 150 mM KOH, 20 mM Tris-base, and incubating for 10 min at 90 °C, as previously described [60]. The solution was then neutralized by adding ~40 μL of 150 mM HCl.

The PCR reactions were multiplexed with the Illumina PCR reverse primers RPI1, RPI2, and RPI4 for cells grown under standard conditions and with RPI9, RPI10, and RPI12 primers for cells subjected to glucose-depletion conditions (Appendix A). To reduce potential PCR artifacts, each cDNA reaction was divided into 7–8 tubes of 10 μL. The PCR reaction mix contained 1 unit of Phusion High-Fidelity DNA polymerase (New England Biolabs), HF Phusion buffer, 0.2 mM dNTP, 0.5 pmol of the respective gene-specific reverse and forward primers (Appendix A) in a final volume of 50 μL, and PCR reactions were carried out for 25 cycles in a Bio-Rad T100 Thermal Cycler. The PCR products were combined and concentrated with a NucleoSpin Gel and PCR Clean-up kit (Macherey-Nagel) according to the manufacturer’s instructions and eluted in 50 μL of Elution buffer. SDS loading buffer was added and the samples were electrophoretically separated on a 6% acrylamide gel containing ethidium bromide. Material above the ligated primers (~120 bp) up to ~200 bp was recovered, and the cDNA was extracted by pulverizing the gel and incubating overnight with mixing at 4 °C in a solution containing 500 mM ammonium acetate, 0.1% SDS, and 0.1 mM EDTA. The supernatant was recovered by centrifugation with Poly-Prep Chromatography Columns (Bio-Rad) to remove the polyacrylamide gel debris; 40 μg/mL of glycogen was added and then 2.5 volumes of 100% ethanol was added and the cDNA precipitated overnight at −20 °C. The cDNAs were collected by centrifugation for 20 min at 4 °C followed by two washes with cold 70% ethanol. The cDNA pellets were dried, resuspended in 14 μL nuclease-free water, and submitted to high-throughput sequencing using Illuminia NextSeq sequencing at the Next Generation Sequencing (NGS) core facility at the Université Paris-Saclay/CNRS (https://www.i2bc.paris-saclay.fr/spip.php?article399&lang=en; accessed on 29 January 2020).

### 2.6. Processing and Analysis of Sequencing Data

The sequencing data were initially processed by the sequencing platform for both the transcriptome samples and the qtPAR-CLIP samples. The Illumina adapters were de-multiplexed using the bcl2fastq2 tool (version 2.18.12) and then removed with Cutadapt (adapter trimming, version 1.15). The quality of the reads was verified using the FastQC tool (version v0.11.5). The sequencing platform subsequently provided a fastq file (including raw sequencing data), a fastq file after trimming (including sequencing data without adapters, ready to be subjected to bioinformatics analysis), as well as a fastqc file containing information on the quality of the sequencing.

The fastq files from the sequencing facility were analyzed on an in-house server. For this, the yeast *S. cerevisiae* (S288C) genome was retrieved from the Saccharomyces Genome Database (SGD; http://www.yeastgenome.org/; accessed on 29 January 2020). This file in the fasta format was indexed (index.fasta) to create a set of "chapters" that facilitated alignments. We used the BWA tool ([61]; version 0.7.12) to perform a local alignment with the following parameters: bwa mem -t 40 –c 1000 index .fasta read-file.fastq > sample.sam. This initial processing generated .sam files, where the reads were aligned to the indexed genome in the same order as the .fastq file, i.e., in disorder. Subsequently, this file was converted to another format that sorted it according to the position on the genome using the Samtools tool ([62]; version 1.9) according to the following parameter: samtools view –bt index.fasta sample.sam -o sample.bam. The PCR duplications were eliminated using the following parameter: samtools rmdup –s sample.bam –o sample-rmdup.bam. This file was then sorted by genomic coordinates using the yeast genome in GFF format (general feature format) available on SGD (genome.gff). This is a tab-delimited text file that describes the genomic characteristics (mRNA, rRNA, tRNA, etcetera) of the reference genome. The sorting was done using the following parameters: samtools sort sample-rmdup.bam > samplermdup-sorted.bam and samtools index sample-rmdup-sorted.bam. The reads were visualized using the IGV software [63]. A table of the read counts was obtained using the Bedtools tool ([64]; version 2.3.0) according to the following parameters: bedtools multicov –S –D –bams samplermdup-sorted.bam –bed genome.gff > sens-counts.txt and bedtools multicov –s –D –bams sample-rmdup-sorted.bam –bed genome.gff > antisens-counts.txt.

The triplicate samples all showed a similar distribution of genes and relative reads, but to minimize read variability between samples, we used the means between samples with nearly the same overall read count in our subsequent analyses. This moderated reads that were aberrantly high or low between samples. Similarly, we used the mean values of the three transcriptome reads. Data were analyzed as above except the PCR-duplication-elimination step was not necessary.

The number of reads obtained by the qtPAR-CLIP technique depended on the abundance of RNA at the time of the experiment. To reduce the distortion introduced through sequencing techniques (RNA lengths and depth of sequencing of a sample), a normalization was carried out using the tables of counting. The RPKM (reads per kilobase of transcript per million mapped reads) was calculated to estimate the abundance of RNAs where the number of reads of each gene was normalized by the length of the gene as well as the total number of mapped reads. The same calculation was carried out on the transcriptome reads [65,66].

### 2.7. Differential Expression

The analysis of the differential expression was made from the raw read counts of the samples of modified PAR-CLIP using package DESeq2 (version 1.14.1) executable on R with a standard of the adjusted P-value of 0.5 and a fold-change of more than 2–fold as previously described [67,68]. Following the analysis, we obtained a list of differentially expressed genes between the two conditions.

### 2.8. Polysome Analyses

Cells (BY4742) were grown to an OD_600_ of between 0.6 and 0.7 mL^−1^cm^−1^ in YPD (yeast extract peptone dextrose) medium with shaking at 30 °C. For the glucose-depletion conditions, cells were collected by centrifugation for 5 min at 5000 rpm in a JA-10 rotor and Beckman Coulter Avanti J-E centrifuge. Cells were then resuspended in medium lacking glucose and grown an additional 5 min at 30 °C. Cultures were harvested by first making the cultures 100 μg/mL in cycloheximide (Sigma), incubating 2 min at 30 °C, and then quickly cooling in an ice bath for 5 min. Cells were then collected by centrifugation, washed twice with lysis buffer containing cOmplete-EDTA protease inhibitor cocktail, 2 mg/mL Pefabloc SC (AEBSF; Roche), and 100 μg/mL of cycloheximide. Collected cells were quick-frozen in liquid nitrogen and stored at −80 °C until needed.

Cells were lysed as described above with lysis buffer containing protease inhibitors and with added cycloheximide, and the absorption at 260 nm was determined with a NanoVue spectrophotometer (GE Healthcare, Chicago, IL, USA). The equivalent of ~250 μg RNA was loaded on an 11 mL 7–47% sucrose gradient with 100 μg/mL cycloheximide, centrifuged for 3 h at 39,000 rpm at 4 °C in a Beckman Coulter Optima L-90K ultracentrifuge and SW41 rotor. In some experiments, samples were incubated for 5 min with 5 mM adenosine 5′-(β,ϒ-imido)triphosphate (AMP-PNP; Sigma) and guanosine 5′-(β,γ-imido)triphosphate (GMP-PNP; Sigma) on ice prior to loading on the gradients.

Twelve-drop fractions (~0.6 mL) were collected with a Retriever 500 (ISCO) fraction collector and a Type 11 Optical unit (ISCO) with 254-nm filters. Samples were made 200 μg/mL in sodium deoxycholate (Sigma), created 50% in trichloroacetic acid (TCA), and then incubated overnight at 4 °C with mixing. The precipitates were collected by centrifuging the samples for 30 min at 13,200 rpm in an Eppendorf 5415R at 4 °C, and they were washed twice with 600 μL acetone and then dried. Samples were resuspended in 2×-concentrated Laemmli loading buffer (Bio-Rad), separated on SDS Laemmli gels, transferred to nitrocellulose membranes, and then visualized with various IgGs. The Thermoscientific (Asnières-sur-Seine, France) PageRuler pre-stained protein ladder was used to determine the approximate molecular weights.

### 2.9. Fluorescence Microscopy

We used temperature-sensitive (ts) mutants of the Sec61 and Sec62 proteins that form the translocon pore in the ER for the import of SRP-dependent polypeptides during translation [55,69]. At the non-permissive temperature, these mutants block or disrupt the Sec61 channel and ER-associated translation is terminated. As a marker of the ER, we used the integrated red-fluorescent-tagged amino terminal domain of Kar2 fused to the HDEL ER retention signal (YIPlac204TKC-DsRed-Express2-HDEL; Addgene, Watertown, MA, USA) in the two *sec* strains; Kar2 is an ATPase that functions as a protein chaperone for refolding proteins within the lumen of the ER ([70] and reference therein), and, consequently, the Kar2 chimera serves as a marker of the ER lumen.

Fluorescence microscopy was performed using an inverted Zeiss Axio Observer.Z1 microscope (Carl Zeiss S.A.S., Oberkochen, Baden-Württemberg, Germany) with a 63 power oil immersion objective equipped with the following filter sets: FITC (filter set 10-Alexa 489, Excitation BP 450/490, Beam Splitter FT 510, Emission BP 515-565) for GFP, and HC (filter set F36-508 Chroma, Excitation BP 562-40, Beam Splitter HC-BS 593, Emission BP 641/75) for RFP. Cell shapes were visualized with DIC (Differential Interference Contrast) optics. Images were acquired with an ORCA-Flash 4.0 charge-coupled device camera (Hamamatsu, Massy, France). Images were treated and analyzed with ZEN 3.1 (Blue edition), and final images were created with Adobe Photoshop 5. No gamma adjustments were made to any image.

## 3. Results

### 3.1. Development of the qtPAR-CLIP Technique

#### 3.1.1. The 4-thioU and UV-A Light Yield Crosslinks under Physiological Conditions

We had previously found that proteins are often unstable when exposed to short UV-C light centered at 254 nm [53]. Moreover, UV-C light generates reactive hydroxyl radicals, and it is often used to sterilize drinking water by efficiently killing cells and destroying viruses [49,71]; thus, these reaction conditions may quickly become non-physiological. In contrast, we expected that cells exposed to UV-A light centered at 365 nm would be unaffected. We tested this by irradiating yeast cells for increasing times with UV-A and UV-C light, with the former also tested after the incorporation of 4-thioU, which is photoactivated by UV light around ~330–340 nm [52]. We used the polystyrene lid for the UV-A light to filter out extraneous light below 290 nm [53]. Our results showed a total abolition of growth of the yeast cells within even a 10 s exposure of UV-C light, while they were unaffected by exposure to UV-A light even after a 30 min exposure under our experimental conditions (Figure 1A). Cells with ~1.3% 4-thioU incorporated (see Materials and Methods) showed a slight decrease in cell viability after irradiation with UV-A (Figure 1A).

To further clarify these results, we created lysates of the irradiated cells, separated the material by SDS-PAGE, transferred to nitrocellulose membranes, and probed with IgG against Ded1. As a control, we used IgG against 3-phosphoglycerate kinase (PGK1). The results showed that Ded1 rapidly disappears when irradiated with UV-C but remained constant with UV-A (Figure 1B). Likewise, PGK1 showed a slow, time-dependent degradation with UV-C but not with UV-A. In contrast, 4-thioU-incorporated cells irradiated with UV-A showed a time-dependent formation of slower migrating species of Ded1, while PGK1, which has no affinity for RNA, was unchanged (Figure 1B).

It was possible that the rapid disappearance of Ded1 in the UV-C irradiated cells was due to the efficient formation of higher-molecular-weight species that did not migrate into the gel rather than due to protein degradation. To test this, we irradiated purified Ded1 protein under the same conditions. We found that Ded1 was rapidly degraded within minutes when irradiated with UV-C but remained stable with UV-A (Figure 1C).

We further studied whether incorporation of 4-thioU would perturb the cell growth significantly. We incubated yeast cells with various concentrations of 4-thiouracil and followed their growth rates; 4-thiouracil was used because it is more efficiently incorporated into yeast cells, and it is readily converted into the nucleotide [57]. The results showed a slight concentration-dependent reduction in growth (Appendix A). Consequently, we concluded that our experimental conditions with 4-thioU and UV-A light were largely benign over the time course of the experiments. Thus, these results enabled us to optimize the irradiation times with UV-A to obtain a high level of Ded1 crosslinking while minimizing the adverse affects on the cells.

#### 3.1.2. Optimization of the qtPAR-CLIP Conditions

Our experimental conditions were based on those previously described for the PAR-CLIP technique, but we streamlined and simplified the protocol to reduce the number of steps and to enhance the recovery of the crosslinked protein [51,72]. Cells were irradiated twice for 10 min on ice with UV-A lamps and with mixing between irradiations, which yielded ~50% of the Ded1 crosslinked as slower migrating species (Appendix A). We used polyclonal antibodies against Ded1 attached to Sepharose-A beads in subsequent steps because we were able to recover nearly 100% of the Ded1 from cell extracts even after stringent washing (Appendix A). We recovered very little material with the Protein A Sepharose beads using the pre-immune IgG (Appendix A). Notably, we used urea and heparin in the washing buffer to minimize nonspecific interactions of proteins and nucleic acids. As previously described for the infrared-CLIP technique [73], we directly digested the IgG-bound complexes with S1 nuclease, which leaves 3′ hydroxyls; this enabled us to eliminate the phosphatase and kinase steps of the original protocol. Thus, the first oligonucleotide adapter was attached on the 3′ end of the RNA directly after nuclease S1 digestion and while the RNA–protein complex was still attached to the IgG Protein A Sepharose beads. Further, this eliminated the effects of zinc-dependent proteases (Appendix A; [74]). As noted by others, this also greatly reduced adapter oligonucleotide oligomerization [75]. A schematic representation of our qtPAR-CLIP protocol is shown in Appendix A and summarized in Appendix A.

### 3.2. Ded1 Crosslinks to Specific RNAs Often at Specific Sites

#### 3.2.1. Ded1 Crosslinks Mostly to mRNAs

Under standard growth conditions (SD-LEU, “WT”), about 70.0% of the RNA crosslinked to Ded1 was mRNAs and ~25.6% on rRNAs, which was mostly 18S rRNA (Figure 2A; Appendix A). About 2.1% of the recovered RNA was retrotransposon RNAs (Gag–Pol) and ~1.2% tRNAs. A small amount of small nuclear RNA (0.8% snRNAs) and noncoding RNA (~0.17% ncRNAs) was also recovered. Only ~0.12% was mitochondrial-derived RNA (mtRNAs); since Ded1 has only been found in the nucleus and cytoplasm, this provided evidence that the crosslinks represented interactions involving intact, nondisrupted cells. The crosslinks on the mRNAs and rRNAs were consistent with Ded1 interacting with the RNAs during translation. However, it is known that Ded1 interacts with mRNAs in the nucleus and in cellular foci (P-bodies and SG) as well.

To test this, we conducted crosslinking experiments with cells that were grown briefly in medium lacking glucose (∆Glu). In contrast to other metabolites or stress conditions, translation initiation is rapidly attenuated within minutes in the absence of glucose without significantly altering the mRNA abundance [76,77,78]. The results showed a ~50% decrease in mRNAs crosslinked to Ded1 and corresponding ~two-fold increase in crosslinks on rRNAs (Figure 2B; Appendix A). However, we also obtained a nearly 10-fold increase in tRNAs crosslinked to Ded1 and nearly a 70% increase in retrotransposon RNAs to ~3.6% of the total. However, the tRNA and retrotransposon results were over-represented because they included some duplicate genes or fragments with the same sequence. Nevertheless, by taking into consideration individual genes, the results indicated at least a greater than three-fold increase in tRNA fragments recovered that were crosslinked to Ded1. Others have similarly found increased Ded1 crosslinks to tRNAs under glucose-depletion conditions [40]. In contrast, we considered only one of the two 35S rRNA gene clusters. There was a slight increase in snRNAs (~0.94%) and a doubling of ncRNAs (~0.35%). Less than 0.04% of the crosslinked RNA was on mtRNAs. These results indicated that Ded1 was participating in translation initiation and translation. The residual crosslinks on the mRNAs in the glucose-depletion conditions could represent mRNAs that were still undergoing translation elongation or the translation of mRNAs not regulated by the glucose signaling pathway. It is also known that translation-inactive mRNAs are quickly sequestered in cellular foci under glucose-depletion conditions [78].

The recovered crosslinked fragments for snRNAs (~0.8%) were largely concentrated within the C/D-box, small, nucleolar RNAs (snoRNA), and to a lesser extent the H/ACA-box snoRNAs (Appendix A). These snoRNAs mostly function as templates for 2′-O-methylation of rRNAs (reviewed by [79]). Thus, these results indicated that Ded1 could be interacting with ribosome precursors within the nucleolus. There was a slightly increased recovery of fragments under the glucose-depletion conditions (~0.94%) and a somewhat altered profile (Appendix A). Interestingly, there was a significant increase in crosslinked fragments of snR17, also known as U3, that is involved in processing the 35S precursor rRNA.

The relatively few crosslinked fragments recovered for the ncRNAs were notable because they were concentrated on SCR1 and *SRG1* (Appendix A). SCR1 is the RNA component of the signal recognition particle (SRP) that is involved in docking paused ribosomes on the Sec61 pore complex of the endoplasmic reticulum (ER) during translation elongation of polypeptides imported into the ER (reviewed by [80,81]). The more than four-fold increase in crosslinks under the glucose-depletion conditions was consistent with Ded1 interacting with ribosome-associated complexes during translation as we expected an increased expression of vacuolar proteins under these conditions. Consistent with this result, we found that Ded1 has high binding affinity for SCR1 RNA in vitro, and the RNA-dependent ATPase activity of Ded1 is inhibited by SRP21 [82].

In contrast, we obtained fewer crosslinked fragments of *SRG1* under the glucose-depletion conditions (Appendix A). SRG1 negatively regulates the transcription of the *SER3* gene through a transcription-interference mechanism; under glucose depletion, the SNF1 kinase complex inhibits the activity of certain transcriptional repressors, which could account for the reduced crosslinking on *SRG1* (reviewed by [76,83]). The three characterized noncoding *SRG1* transcripts are capped and polyadenylated, and they can be exported to the cytoplasm, where they are eventually degraded [84]. The largest transcript contains the *SER3* gene, and it can undergo translation initiation and subsequently nonsense-mediated decay (NMD). Therefore, it was possible that the crosslinks resulted from Ded1-associated RNAs undergoing translation. However, the recovered crosslinked fragments under standard growth conditions all mapped at ~120 residues from the 5′ end of the transcripts and consequently were in common with all three transcript products.

#### 3.2.2. Glucose Depletion Redistributes the Crosslinks on rRNAs

The large increase in the percentage of crosslinks on rRNAs under the glucose-depletion conditions was somewhat surprising. Ribosomal RNAs are often recovered in CLIP and CRAC experiments of RNA binding proteins because of their high abundance; it was possible that Ded1 freed from the mRNAs might randomly interact with ribosomes. However, it also could be because Ded1 was sequestered with ribosome-bound mRNAs that were paused during translation or that were present in cellular foci. Under standard conditions, nearly 60% of the rRNA crosslinks were on the 18S rRNA and about 35% were on 25S rRNA (Figure 2C). Only ~3.2% and ~0.6 were on 5.8S and 5S rRNAs, respectively. Less than 2% of the recovered crosslinked fragments mapped to the external transcribed spacers (ETS) and internal transcribed spacers (ITS) that are found in the 35S precursor rRNA. Thus, Ded1 was interacting primarily with mature ribosomes. Under glucose-depletion conditions, the percentage of crosslinks on the 18S and 25S rRNAs was reversed, but they were proportional to the total quantity of RNA (residues) of each species present in the 80S ribosomes (32.9% and 62.0%, respectively; Figure 2D). Similarly, the 5.8S (~3.8%) and 5S (~1.0%) crosslinks more closely approximated their relative abundance within the ribosomes (2.9% and 2.2%, respectively). The ETS and ITS crosslinks were similar to the standard conditions. Thus, these results showed Ded1 interacting primarily with the 40S (or complexes thereof) ribosomal subunit(s) during translation under standard conditions and with the 80S ribosomes under glucose-depletion conditions where translation initiation was largely inhibited. The latter result could represent 80S ribosomes, perhaps stalled, on the translating mRNAs or interactions with free 80S ribosomes not bound to mRNAs; 80S monosomes are known to increase under glucose-depletion conditions and also to form non-translating, free 80S ribosomes [76,85]. However, the results did not support crosslinking of ribosomes in cellular foci because only the 48S PIC involved in scanning the mRNAs is thought to be sequestered within these foci (reviewed by [31]).

#### 3.2.3. Crosslinks Partially Correlated with mRNA Abundance

Under standard conditions, 70% of the crosslinked fragments that we recovered were for mRNAs (Figure 2A). However, the vast majority of the identified mRNAs had relatively few crosslinks (Appendix A). Indeed, about half of the mRNAs showed no crosslinks at all, while only a few hundred showed high levels of crosslinking. This was nevertheless expected as only a subset of the ~6000 yeast genes are expressed at any given time (reviewed by [86]). This was even more pronounced under the glucose-depletion conditions, which indicated that Ded1 was interacting with fewer mRNAs under the repressed translation (∆Glu) conditions (Appendix A). Thus, Ded1 seemed to be interacting primarily with specific mRNAs, but it was also possible that the Ded1 interactions reflected the mRNA abundance and accessibility. Highly expressed and long mRNAs might show more crosslinks because the probability of Ded1 interacting with those mRNAs would be higher. Further, certain mRNAs encoding abundant proteins might be undergoing a high level of translation that facilitated the interactions (and crosslinks) with Ded1.

Ded1 is an ATP-dependent RNA binding protein. Thus, if Ded1 was randomly interacting with mRNAs, then we would expect a strong correlation with the mRNA abundance (size and quantity of mRNA transcripts). We plotted the normalized frequency of recovered crosslinked genes against the normalized transcriptome based on the mRNA transcript size and abundance (reads per kilobase of exon per million reads mapped or RPKM; Figure 3A). As expected, about half the transcriptome mRNAs showed no crosslinks regardless of their abundance. Nevertheless, there was a clear correlation with the mRNA abundance for the mRNAs that were crosslinked. However, this correlation became much less pronounced under the glucose-depletion conditions (Figure 3B), and it further decreased when only the most highly crosslinked mRNAs were taken into consideration (Appendix A). These results indicated that additional parameters were involved in the crosslinking frequency.

#### 3.2.4. Ded1 Was Highly Crosslinked to Few mRNAs

Our results indicated that Ded1 was interacting primarily with a subset of mRNAs undergoing active translation. Under standard conditions, the recovered mRNAs with the most crosslinked fragments often corresponded to translation initiation or translation elongation factors (e.g., *TEF2*, *YEF3*, *EFT2*, *TIF4631*, *DED1*) and ribosomal proteins (e.g., *RPS5*, *RPL10*, *RPL31A*, *RPS20*, *RPP0*, *RPS3*, *RPS31*; Appendix A). Others have similarly found that Ded1 and DDX3 crosslink particularly to mRNAs encoding ribosomal and translation factors [40,87]. This result was partially expected as yeast exerts a majority of its translational activity in the production of ribosomal proteins under active growth conditions (reviewed by [88]). However, only certain mRNAs of this subset of mRNAs that were expected to be highly translated were recovered, which indicated that Ded1 was crosslinking preferentially with only a small fraction of the potential mRNAs (~200 ribosomal RNAs). Thus, Ded1 might serve a potential regulatory role. Further evidence for this was that Ded1 was highly crosslinked to its own mRNA; that is, Ded1 may also regulate its own expression, as previously proposed [40]. Other prominently crosslinked mRNA fragments corresponded to mRNAs encoding various metabolic proteins, particularly involved in glycolysis and sugar metabolism (e.g., *TDH3*, *CDC19*, *FBA1*, *ITR1*, *PDC1*, *ENO1*, *FKS1*, *PGK1*, *PGI1*). Therefore, Ded1 was associated with a subset of mRNAs undergoing high levels of translation, and this underlined the risks of using Ded1 mutants to characterize its role in the cell.

In contrast, under glucose-depletion conditions, we obtained a significantly different profile for the most highly crosslinked mRNAs (Appendix A). The translational factors and ribosomal protein mRNAs were replaced by those involved in metabolic and physiological changes. For example, we recovered mRNAs encoding cell wall, plasma membrane, mitochondrial, and vacuolar proteins (e.g., *TAR1*, *PIL1*, *CCW12*, *GUT2*, *PMA1*, *APE3*, *OM45*, *PRB1*, *CTS1*). However, *TAR1* is encoded on the antisense strand of 25S rRNA, and, consequently, it may be over-represented in the listing of upregulated mRNAs. Others reflected the altered metabolic pathways and response to stress (e.g., *GLK1*, *TDH3*, *PDC*1, *PGK1*, *TPS1*, *FBA1*, *CMK2*, *HXK1*, *DDR2*). Interestingly, eight mRNAs were highly crosslinked under both conditions, which included the translation factors *TEF2* (plus *TEF1*) and *DED1* (Appendix A). The mRNAs of *PDC1*, *HXK1,* and *PGK1* are thought to go to P-bodies and SG under glucose-depletion conditions in yeast (reviewed by [77]). Thus, it was possible that the crosslinks also reflected mRNAs stored in cellular foci. However, others have found that *HXK1* and *GLK1* genes are repressed by Hxk2 in the presence of glucose and highly expressed under glucose-depletion conditions, which would be more coherent with the active translation of these mRNAs [89].

#### 3.2.5. Glucose-Depletion Crosslinks Reflected Altered Genetic Expression

Glucose depletion rapidly alters the translational landscape within minutes [76,77,78]. We compared the crosslinking pattern of cells grown under standard and glucose-depletion conditions using the DESeq2 package [67]. In contrast to the previous analyses, this program uses shrinkage estimation for dispersions (variance between replicates) and fold changes to determine the differential crosslinking patterns under the two conditions. The results are shown in Appendix A.

The results were largely consistent with the overall frequency of the recovered crosslinked fragments under the two conditions, but additional elements were revealed. We subjected the 20 most upregulated mRNAs (Table 1) and 20 most downregulated mRNAs (Table 2) to a Gene Ontology (GO) Slim Term Mapper analysis for the process of the encoded protein using the Saccharomyces Genome database website (https://www.yeastgenome.org/goSlimMapper; accessed 12 May 2021). The results showed a downregulation of mRNAs encoding ribosomal factors and an upregulation of mRNA encoding proteins involved in stress and in a metabolic shift. This indicated that Ded1 was crosslinked to actively translated mRNAs.

If Ded1 was randomly interacting with the mRNAs, then we would expect a random distribution of the crosslinked fragments, but this did not appear to be the case (Figure 4). Some mRNAs showed a broad distribution of fragments (e.g., *YEF3*, *PMA1*, *GLK1*), while others showed very pronounced peaks at specific sites (e.g., *SRP40*, *TDH3*, *TIF4631*). Notably, few fragments corresponded to intron sequences (e.g., *RPL31A*), indicating that Ded1 was crosslinking to processed mRNAs that were most likely cytoplasmic mRNAs undergoing translation. Some prominent peaks appeared on highly expressed mRNAs (e.g., *TDH3*), while others were on weakly expressed mRNA (e.g., *SRP40*). Thus, it did not appear that the Ded1 crosslinks on the mRNAs were random interactions. The Ded1 mRNA was one of the few mRNAs that showed significant crosslinks on the 3′ UTR. Yeasts typically have short 5′ and 3′ UTRs, where the mean length and standard deviations for the 5′ UTR, ORF and 3′ UTR are 96.5 ± 116.8, 1537.1 ± 1164.8, and 146.7 ± 138.4, respectively, although the polyadenylation cleavage site can vary for the 3′ UTRs [90,91]. The mRNA encoding Ded1 is unusual because it has a short 5′ UTR of 21 residues and a long 3′ UTR of up to ~722 residues. The crosslinks on the *DED1* mRNA were most clustered on the 5′ half of the ORF, but a strong signal was obtained centered at ~415 residues after the stop codon (Figure 4, Appendix A). Similar results were previously reported using the CRAC method of crosslinking [40]. The 3′ UTRs often regulate mRNA localization, mRNA stability, and translation (reviewed by [92]). Thus, it was possible that Ded1 was auto-regulating its own expression, as previously proposed [40].

In contrast, *GCN4* has a particularly long 5′ UTR of 571 residues (Figure 4, Appendix A). However, this 5′ UTR is unusual because it contains four upstream ORFs (uORFs) that regulate the translation of the *GCN4* mRNA (reviewed by [46]). Translation and termination of the uORFs inhibits re-initiation and scanning of the 48S PIC to the AUG of the *GCN4* ORF. In contrast, under amino acid starvation conditions, the scanning distance and time required to re-initiate on the mRNA are increased, and the Gcn4 protein is expressed. The fourth uORF plays a particularly important role in this regulation, and the peak fragments of crosslinks to Ded1 were centered at uORFs 3 and 4 under standard conditions but greatly reduced under glucose-depletion conditions (Figure 4, Appendix A). Thus, these results indicated that the crosslinks between Ded1 and the uORFs formed during translation. Similar results have been noted by others [40]. Interestingly, the 5′ UTR of *GCN4* is uridine-rich (~35%), as is the 3′ UTR of *DED1* (~47%), although the crosslink peak(s) occur(s) at (a) specific site(s) that do not reflect the density distribution of uridines.

#### 3.2.6. Crosslinks Poorly Correlated with mRNAs UTR Length

Ded1 is an ATP-dependent RNA helicase that is thought to be particularly important for unwinding RNA secondary structures in the 5′ UTRs [34,40,41,42,43,44]. However, the 5′ UTRs of yeast, in general, are A-rich and predicted to be largely unstructured [94,95]. We recovered very few crosslinked fragments that mapped to the 5′ UTRs, which was in contrast to the work of others using the CRAC UV-C crosslinking technology [40,41]. Nevertheless, we previously showed that Ded1 interacts with the eIF4F complex in a manner independent of RNA [19]; thus, it was possible that the interactions with the 48S PIC were such that reactive functionalities were not in close proximity, whereas, during translation elongation, the interactions would be better positioned. Therefore, the absence of crosslinks would not be indicative of an absence of interactions. Nevertheless, we expected that the more highly crosslinked mRNAs overall would represent the preferential substrates for Ded1.

We plotted the RPKM values for the crosslinked mRNAs against the 5′ and 3′ UTR lengths under standard and glucose-depletion conditions (Appendix A). In general, the majority of the recovered mRNAs had UTRs lengths that were within the expected mean range, which was also true for the ORFs. However, there were some exceptions. For the mRNAs with the highest recovery of crosslinked fragments (shown in red in Appendix A), *GCN4*, *RPS26A*, *RPS8B*, *URA2*, *ACC1,* and *SIM1* had particularly long 5′ UTRs. The importance of the 5′ UTR length was less evident under the glucose-depletion conditions.

For the 3′ UTR, *DED1* and *PMA1* were prominent, as was *HAC1* in the case of glucose-depletion conditions. Interestingly, translation of *HAC1* is negatively controlled by an interaction between the unspliced intron and 5′ UTR under standard conditions, which is reversed by noncanonical splicing of the intron by the RNase Ire1 under stress conditions [96]. No crosslinked fragments were isolated for the intron sequence, which provided further evidence that the crosslinks observed on Ded1 reflected mRNAs undergoing active translation. The uncharacterized gene *YLR154C-G* was found in all cases, but this was probably an artifact because it overlaps with the RDN37-2 rRNA genes, and, consequently, the reads cannot be attributed uniquely to *YLR154C-G*.

#### 3.2.7. Ded1 Preferentially Binds Purine-Rich Sequences

We used the MEME server (https://meme-suite.org/meme/tools/meme; accessed on 24 January 2022; [97,98]) with different settings and parameters to determine if Ded1 was preferentially binding certain sequences or motifs. To facilitate these analyses, we used regions with 20 or more unique reads and included 10–20 residues of either side in case the crosslinks were not centered on the motif region. We expected a clear bias towards regions that were uridine-rich because these regions would have the highest proportion of incorporated 4-thioU. This was the case for certain fragments; for example, the 843-residue-long 3′ UTR of *DED*1 contains 46.7% uridines, although, oddly, the fragment containing the peak of crosslinked reads was not in the most uridine-rich regions. Surprisingly, our MEME analyses found that the majority of the peaks within the ORFs were instead in purine-rich sequences, but the actual sequences identified were highly variable with no clear consensus. An example of these analyses is shown in Appendix A. In general, the best fit was a weak consensus sequence of R-U-Y-G-A-A, where R is a purine and Y a pyrimidine. Although these results were unexpected, we and others have found that the RNA-dependent ATPase activity of Ded1 is most stimulated by poly(A) RNA [54]. Similarly, others have found that DDX3 crosslinks preferentially to purine-rich sequences [87]. Thus, crosslinking studies with incorporated 6-thioguanosine might be more revealing of any potential motif.

We also analyzed the sequences containing the read peaks for any secondary or pseudoknot structures using the RNA Structure web server (https://rna.urmc.rochester.edu/RNAstructureWeb/; accessed on 17 March 2022; [99]). Unsurprisingly for purine-rich sequences, no prominent structures with Watson–Crick or G–U base pairings were found. Thus, the context of Ded1 relative to other cofactors probably plays a determining role in crosslinking sites.3.2.8. Ded1 Crosslinks on Retrotransposon RNAs

Ded1 has previously been identified as being important for the replication of the double-stranded RNA virus L–A [25]. This virus encodes two ORFs that overlap by 130 residues: an encapsulation protein (Gag) and a fusion protein containing an RNA-dependent RNA polymerase (Gag–Pol), the latter of which results from a −1 ribosomal frameshift that occurs in about 2% of the translation events (reviewed by [100]). Although unrelated, the yeast genome contains five classes of Ty long-terminal-repeat (LTR) retrotransposons (Ty1–Ty5), with numerous variants within each class, that have a similar mode of expression, although, in this case, the Gag–Pol fusion product is formed from a +1 ribosome frameshift at a 3–13% efficiency (reviewed by [101,102]). Thus, it was not surprising that we obtained significant crosslinking between Ded1 and the retrotransposon RNAs (Figure 1; Appendix A). Two examples are shown in Appendix A.

Although there are multiple variants, many of these elements encode regions with the same or similar sequence, and, consequently, the retrotransposons may be over-represented in our analyses. Nevertheless, under standard conditions, only crosslinked fragments from a few retrotransposons were recovered that were mostly different Ty1 (Ty1-1–Ty1-5) elements, which reflected their relative abundance [101]. We recovered a higher percentage of crosslinked fragments overall under the glucose-depletion condition, but these crosslinks were more uniformly distributed over the different retrotransposon RNAs, which roughly corresponded to their overall abundance in the genome (~3.5%), but they under-represented the overall poly(A)-containing RNA level of the retrotransposons (5–10%; [101,102]). However, only a small percentage of these poly(A)-containing retrotransposon RNAs undergo translation at any given time. Crosslinks were found within the ORFs and not in the LTRs, which suggested that Ded1 interacted with retrotransposon RNAs undergoing translation. The Ty1 and Ty3 retrotransposon RNAs also form cytoplasmic foci, T-bodies, or retrosomes that are similar to virus-like particles (VLPs), so it was possible that crosslinks also occurred within these foci [101,102]. Indeed, the Gag protein is translated into the lumen of the ER through the signal recognition particle (SRP), presumably to prevent premature association of Gag with the Ty1 RNA during translation; the Gag protein subsequently exits the ER by an unknown mechanism and associates with the Ty1 RNA to form RNPs that eventually form the VLPs, where the Ty1 RNA is subsequently reverse transcribed [101,103]. Thus, the VLPs represent Ty1 RNAs that are undergoing active translation. The crosslinks on the retrotransposon RNA and the SCR1 RNA of the SRP implicated Ded1 being involved in this process.

#### 3.2.8. Ded1 Crosslinked to tRNAs

We recovered a significant number of tRNA fragments crosslinked to Ded1, which was greatly increased under the glucose-depletion conditions (Figure 2A,B, Appendix A). This was somewhat surprising because tRNAs form highly compact tertiary structures that lack obvious binding sites for Ded1, although we do find that tRNAs were good substrates for activating the RNA-dependent ATPase activity of Ded1 [104]. The vast majority of the recovered fragments were on tG(GCC), followed by tD(GUC), tS(AGA), and tE(UUC) under standard growth conditions. The recovered fragments were more broadly distributed under the glucose-depletion conditions, but tG(GCC) and tS(AGA) were particularly well crosslinked (Appendix A). Interestingly, both tG(GCC) and tS(AGA) are implicated in translational regulation under stress conditions [105].

We expected S1 nuclease to cleave the tRNAs primarily in the anticodon loop; as a result, we obtained fragments corresponding to the 5′ and 3′ halves of the tRNAs, but most of the fragments corresponded to the 3′ halves (Appendix A). However, there was a bias because the TψC loop is typically uridine-rich, while the D-loop contains dihydrouridines that lack a conjugated ring structure; we did not expect a modified 4-thioU at these positions to be activated by UV-A light under these conditions. Consistent with this, many of the U to C transitions mapped to the acceptor stem and TψC loop (Appendix A). Note that pseudouridines can undergo the U to C transition during reverse transcription, and, consequently, it may be over-represented [106]. These results indicated that Ded1 likely interacted with tRNAs during translation, although aminoacylated tRNAs serve other biological roles as well (reviewed by [107]).

The tRNA halves or fragments thereof (tRFs) are known to serve important regulatory functions (reviewed by [105,107]). In yeast, the 5′ and 3′ tRFs are associated with the ribosomes, particularly under stress conditions, and these interactions result in translational inhibition [105]. Interestingly, these tRFs do not bind the ribosomes in the “classical” A- or P-tRNA sites. The tRFs are generated by Rny1, an RNase T2-like enzyme, that leaves 5′ hydroxyls and 3′ phosphates that would not be substrates for the adapter oligonucleotides used under our experimental conditions. Nevertheless, they may have been further digested by S1 nuclease; therefore, we cannot distinguish between crosslinks on intact tRNAs versus tRFs.

#### 3.2.9. Ded1 Crosslinks on 80S Ribosomes

Under standard conditions, the crosslinks on rRNAs were concentrated on the 18S, which indicated that Ded1 interacted primarily with the 40S (or complexes thereof) ribosomal subunit(s), while they were more uniformly distributed under the glucose-depletion conditions (Appendix A). Similar results were obtained under standard conditions by others [40]. We mapped the peak distribution of recovered crosslinked fragments (as shown in Appendix A) and the positions of U to C transitions on the solved cryoelectron microscope structure of the 80S ribosomes with bound mRNA and tRNA (Figure 5; Appendix A; [108]).

All of the U to C transitions were solvent-exposed in this structure, as were portions of the rRNA peak fragments. The majority of the crosslinks mapped on the 18S rRNA, but U to C transitions were scattered on the surface of the 25S rRNA as well. None of the crosslinks were uniquely associated with the 48S PIC (i.e., covered by the 60S), so we cannot distinguish between 40S crosslinks on the initiation complex and those on the intact ribosome. In contrast to others, we found many of the crosslinks on the lagging strand side of the ribosomes, but residues surrounding the entry site showed extensive crosslinking as well [34,41]. This would suggest that the entry site was blocked by other proteins, such as elongation factors. Notably, the elbow of the bound tRNA was well exposed in this structure, which also suggested that the crosslinks on the tRNAs were from ribosome-bound tRNAs.

The crosslinks did not indicate a single binding site on the 80S ribosomes as they were too widely distributed relative to the size of Ded1. Instead, the results most likely reflected Ded1 binding at various distances on the mRNA and then periodically bouncing back (binding) on the ribosomes like a paddle ball. These results would be consistent with Ded1 binding 3′ to the ribosomes on the mRNAs, as previously proposed [34,41]. Thus, Ded1 would be in a position to resolve RNPs and RNA structures that impede the translocating ribosomes, but the ribosomes themselves can have “helicase” activity, as has been shown for prokaryotic ribosomes (reviewed by [109]). Instead, Ded1 may use its ATP-dependent RNA binding activity to pause translocating ribosomes or to stabilize ribosomes already paused.

**Figure 5 genes-14-01566-f005:**
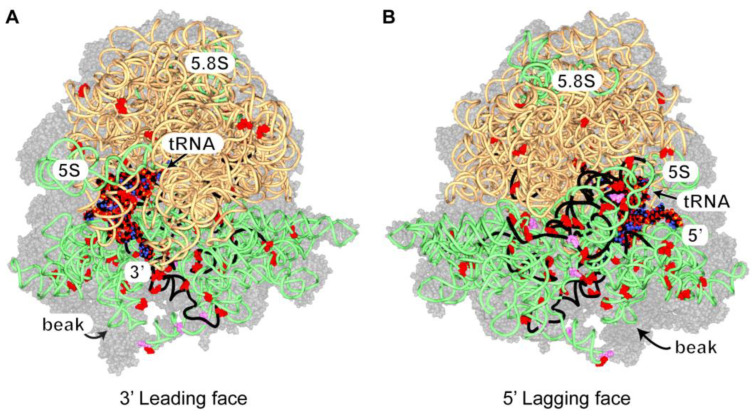
The distribution of Ded1 crosslinks on rRNAs. The 40S and 60S ribosomal proteins are shown with 50% transparency as gray space-filling models; the 18S, 5.8S, and 5S rRNAs are shown as light green spaghetti; the 25S rRNA is beige spaghetti; and the bound tRNA and mRNA are shown as space-filling models. Note that the space-filling model is shown with standard colors that should not be confused with crosslinked sites. The 5′ and 3′ ends of the mRNA are as indicated. The positions of single or multiple U to C transitions are shown as red or magenta, respectively, space-filling residues, and the RNA fragments highly crosslinked to Ded1 (≥60; Appendix A) are shown as black spaghetti. (**A**) The ribosomes from the perspective of the 3′ end of the mRNA. The 40S ribosomal subunit is on the bottom. (**B**) The ribosomes from the perspective of the 5′ end of the mRNA. The ribosome structure with bound SDD1 mRNA and tRNA in the P site (PDB 6snt) was used and viewed with Swiss PDBViewer [108,110].

### 3.3. A Subpopulation of Ded1 Was in Close Proximity to the ER

Our crosslinking results showed that Ded1 interacted with the SCR1 RNA of the SRP, with retrotransposon RNAs and with mRNAs encoding vacuolar proteins; these data indicated that a subpopulation of Ded1 was associated with the ER. However, we and others have previously shown that Ded1 has a relatively diffuse location within the cell under normal growth conditions [19,27,28,29,30,31]. Therefore, we used temperature-sensitive (ts) mutants of the ER translocon proteins Sec61 and Sec62 to block ER-associated translation at the nonpermissive temperatures, and consequently to accumulate any Ded1 on mRNAs encoding ER-targeted proteins. In addition, we used an ATPase-inactivated mutant of Ded1, Ded1–DQAD, that was previously shown to have a high propensity to localize in cellular foci that are no longer undergoing translation [30]. We expected that Ded1 would form foci in proximity of the ER at the nonpermissive temperatures. Our results showed that this indeed was the case for both the *sec61* and *sec62* mutant strains (Appendix A).

We next asked whether the Ded1 foci were in close proximity to the retrotransposon VLPs present near the ER. As for a number of RNA viruses, the retrotransposon RNA is encapsulated by the Gag protein to form the VLPs, whereupon it is reverse transcribed into the double-stranded-DNA Ty1 element [101,102]. Consequently, we used the pLTRp:Gag_1–401_:GFP:ADH1_TER_ plasmid that has the Ty1 U3 promoter, 59 residues of the 5′ UTR, the Ty1 Gag ORF (1–401), and a 3′ GFP(S65T) ORF (a kind gift of Joan Curcio; [103]); this plasmid-borne Gag–GFP fusion construct enabled us to follow Gag expression by fluorescence microscopy.

The results are shown in Figure 6, and they were consistent with the VLP associated with the ER. We co-expressed the Gag–GFP construct with Ded1–mCh, which was subsequently subjected to heat shock at 37 °C, and we found that the VLP and Ded1 foci often appeared to be superimposable (Figure 6). However, a closer examination indicated that they were largely separate foci in close proximity. This was most apparent in time-lapse films (Appendix A). Cells grown at 20 °C showed this effect more clearly because of the slower movement of the foci. Thus, we had no evidence that Ded1 and Gag were in the same foci. Rather, the results indicated that Ded1 associated with Gag mRNA undergoing translation, and with Gag subsequently associating with the translated retrotransposon RNA to form the VLPs. Interestingly, cells grown in the absence of glucose seemed to have smaller VLPs in general, which would be expected with an attenuation of Gag–GFP translation under these conditions.

### 3.4. Ded1 Was Associated Transiently with Polysomes

If Ded1 was associated with translating ribosomes, then we expected it to be found in the polysomes fractions of sucrose gradients. We previously used sucrose gradients to show that the majority of Ded1 sedimented around and above the 40S ribosomal subunits but below free proteins, which indicated that Ded1 was associating with RNAs as RNPs, as might occur with Ded1 bound to the CAP-binding complexes (eIF4F or CBC) on mRNAs that were not actively undergoing translation [19]. Indeed, mass spectrometry analyses of these fractions that were precipitated with IgG against Ded1 recovered most of these cofactors. However, we did find in this previous work that a small amount of Ded1 was present in all the heavier sedimenting fractions as well. Other workers have noted similar results with *Leishmania* and mammalian DDX3 [34,36]. This observation seemed to indicate that only a small percentage of Ded1 associated with translating ribosomes at any given time. An alternative explanation was that most of the Ded1 dissociated from the mRNAs in the polysome fractions during the sample preparation.

We modified our previous protocol to enhance the amount of Ded1 found in the polysomes. We obtained the best results using 2–5 mM MgCl_2_; the bulk of Ded1 sedimented at the very top of the gradients (with free proteins) with 20 mM MgCl_2_, which indicated that Ded1 dissociated from the RNAs. Ded1 is an ATP-dependent RNA binding protein. To further stabilize Ded1 on the mRNAs, we incubated the cell extracts for 5 min on ice, prior to loading on the gradients, with 5 mM AMP-PNP, which is a nonhydrolyzable analog of ATP, to “lock” Ded1 into a conformation with a high affinity for RNAs. Moreover, we added the nonhydrolyzable GTP analog GMP-PNP to further stabilize the complexes as GTP hydrolysis is used during formation of the 48S PIC into the 48S IC, and because associated complexes, such as the SRP, likewise use GTP (reviewed by [80,81]). The results are shown in Figure 7.

We obtained polysome gradients with somewhat better resolved peaks when AMP-PNP and GMP-PNP were present. Notably, we obtained a small side peak that probably corresponded to the 48S PIC/IC. However, the majority of Ded1 sedimented above this peak, while a smaller amount was distributed throughout the heavier sedimenting material, although we did often obtain an increased signal corresponding to the 80S ribosomes as well. Incubating the cells under glucose-depletion conditions for 5 min prior to extracting the cells yielded significantly larger peaks for the 40S and 60S subunits and for the 80S ribosomes and smaller peaks for the polysomes, but the distribution of Ded1 was similar (Appendix A). There may have been a slight shift in the peak distribution to heavier sedimenting regions, but we saw this with some of the standard growth condition profiles as well. These data were consistent with the continued, but reduced, translation of Ded1-associated mRNAs under glucose-depletion conditions. However, the added AMP-PNP and GMP-PNP did not significantly alter the Ded1 distribution pattern. These data supported the hypothesis that the majority of Ded1 dissociated from the polysome fractions during the sample preparation and before the AMP-PNP was added. The interactions of Ded1 with the cap-binding complexes are protein-mediated, and they are independent of ATP; this would explain why it accumulated with RNPs around the 40S peak [19].

## 4. Discussion

Our crosslinking data indicate that yeast Ded1 interacts primarily with mRNAs undergoing translation (~70%) and with rRNAs (~26%). The types of RNAs and relative percentage of crosslink fragments recovered are remarkably similar to those obtained for mammalian DDX3 in FAST-iCLIP crosslinking experiments [87]. As for these authors, we find that Ded1 crosslinks to purine-rich sequences with no obvious consensus sequence. Ded1 is considered a translation initiation factor that facilitates the 48S PIC scanning to the initiation codon and formation of the 48S IC that is competent for association with the 60S ribosomal subunit and formation of the 80S ribosomes involved in elongation [40,41,42,43,44]. Our results presented here show that Ded1 either remains associated with the translational machinery after this point or that it re-associates with the translocating ribosomes during translation. This is further supported with our glucose-depletion conditions that show reduced crosslinking on mRNAs in general and a shift to mRNAs expressing proteins important for the stress response and the metabolic shift.

The crosslinking efficiency of a protein on RNAs depends on the probability that reactive functionalities are in close proximity. In our experiments, this would depend on how often Ded1 is bound near 4-thioU residues and whether reactive functionalities on Ded1 are in close proximity to the activated 4-thioU when bound. This, in part, explains the differences between our experiments with qtPAR-CLIP and those completed with UV-C (CRAC- and CLIP-type). UV-C generates diffusible hydroxyl radicals and consequently more generalized reactivity. In contrast, the excitation of the 4-thioU yields a reactive ππ* triplet state that rapidly decays within microseconds, and where the most reactive amino acids (tyrosine, tryptophan methionine, lysine, and cysteine) would need to be in close proximity (reviewed by [112]). We previously showed that Ded1 interacts with CAP-associated factors (cytoplasmic eIF4F and nuclear CBC) predominantly through protein–protein interactions [19]. This would partially explain why we see very few crosslinks on the 5′ UTRs of mRNAs even if we and others have evidence for their presence. In contrast, we see efficient crosslinking on the mRNA ORFs, which indicates that the binding interactions are different, and that Ded1 is making more direct (or altered) contacts with the RNAs.

By the same criteria, we would have expected that most of the recovered crosslinked fragments would be rich in uridines as the higher density would increase the probability that an activated 4-thioU would be present. However, this is not the case. With relatively few exceptions, the majority of the recovered crosslinked fragments are purine-rich in sequence and consequently contain relatively few 4-thioUs. However, this result is consistent with the enzymatic activity of Ded1 being most stimulated by A-rich sequences [54]. Similarly, others have found that DDX3 preferentially crosslinks to purine-rich sequences [87]. Ded1/DDX3 proteins are known to interact with G-quadruplexes [113,114]. It was possible for Ded1 to crosslink to such complexes under our conditions as the looped regions could contain 4-thioUs, but we did not recover any such fragments, possibly because the protein–RNA interactions were not oriented in a fashion suitable for crosslinking but also because of the rarity of such structures in yeast mRNAs [115].

The eIF4F initiation complex is 5′ to the scanning 48S PIC on the mRNAs [45]. Our crosslinking results show Ded1 interactions close to both the 5′ exit and 3′ entry sites of the mRNA on the 40S/48S ribosomal subunit, which indicates that Ded1 is in close proximity to both sites at different steps during translation initiation and elongation. The laboratory of Tollervey obtained similar results using the CRAC methodology [40], while other workers found that yeast Ded1 and DDX3 are located primarily near the 3′ entry site [34,41]. We actually see somewhat fewer crosslinks in the immediate vicinity of the 3′ entry site, most likely because elongation factors are often present at this position that block Ded1. However, the “elbow” of the tRNAs bound to the ribosomes might be more exposed, which would account for the crosslinked residues (U to C transitions) at this position. Thus, we have no evidence that Ded1 forms specific interactions with the ribosomes; rather, it appears that Ded1 is forming transient interactions with the rRNAs, particularly the 18S rRNA, during initiation and elongation. Although we have no crosslinks that are clearly 48S-specific (i.e., at positions that would be subsequently protected by the 60S ribosomal subunit), these data nevertheless support a model with Ded1 interacting with the eIF4F complex during initiation and subsequently shifting to the 3′ leading strand of the mRNAs. Similar results were obtained by others [40].

Our qtPAR-CLIP crosslinking protocol has very few adverse effects on the cells, and we are able to optimize the crosslinking efficiency under physiological conditions to a high degree. We obtain crosslinked fragments for a large number of different mRNAs, which supports previous work showing Ded1 functioning as a general translation factor [22,41,116]. However, only relatively few of the mRNAs show a high degree of crosslinking, which is often associated with mRNAs encoding proteins involved in translation. It is likely that the translation of these mRNAs is highly regulated and that Ded1 might contribute to this regulation, and, hence, it might bind the RNAs more extensively (longer). Indeed, DDX3 was shown to bind highly expressed mRNAs in general but to affect the expression of only a small subset of these RNAs [34]. Moreover, DDX3 accumulates on the ORFs of mRNAs with stalled ribosomes in cells treated with arsenite [87].

Ribosome pausing is an important regulatory element, and it is associated with co-translational protein folding, protein targeting, mRNA and protein quality control, and with co-translational mRNA decay (reviewed by [117]). The ATP-dependent clamping properties of Ded1 on RNAs may be important in this process by pausing and/or stabilizing the paused ribosomes. Indeed, genetic depletion of *Leishmania* DDX3 leads to impaired elongation of the translating ribosomes, which triggers the co-translational quality-control system of the newly synthesized polypeptides [36]. Conditions that pause ribosomes lead to a large increase in mammalian DDX3 crosslinks to the ORFs of mRNAs, implicating a role for DDX3 in translation elongation [87]. Moreover, while mammalian 5′ UTRs tend to be long and structured, yeast 5′ UTRs are generally short, A-rich, and unstructured [34,94,95]. Thus, the ATP-dependent RNA binding properties of Ded1 may be used primarily to stabilize and/or pause the ribosomes during scanning and elongation, which is further supported by the enhanced crosslinking on SCR1, retrotransposon RNA, and mRNAs encoding vesicle proteins under glucose-depletion conditions. Thus, Ded1 can be considered as an ATP-dependent molecular switch or clamp to help control translation elongation.

## Figures and Tables

**Figure 1 genes-14-01566-f001:**
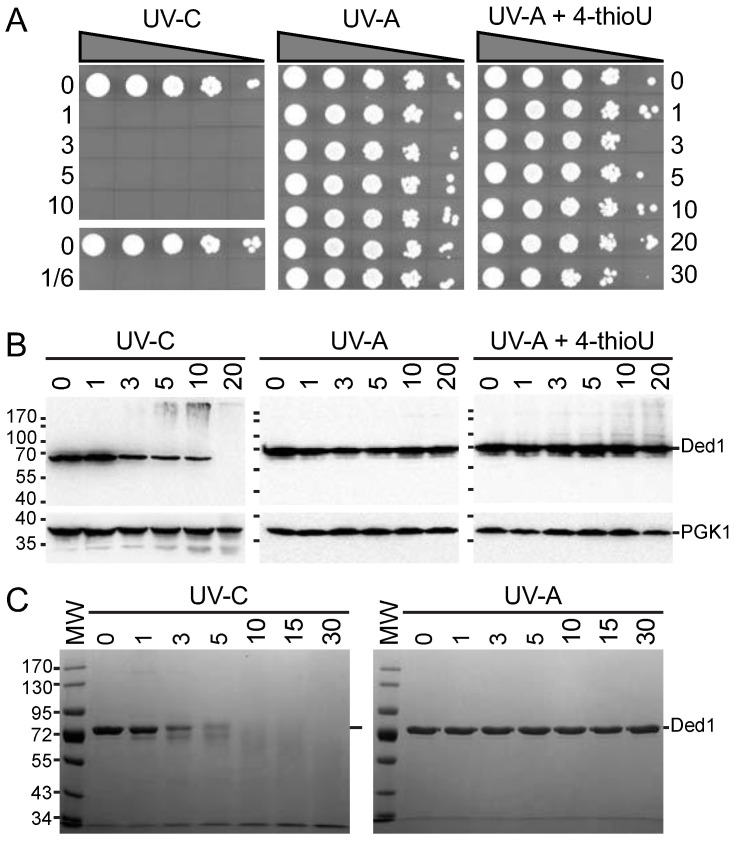
Effects of UV light at 254 nm and 365 nm on cell growth and protein stability. (**A**) Serial dilutions (10-fold) of cells exposed to the indicated UV light source, in the presence and absence of 300 μM 4-thiouracil, for the times shown in minutes spotted on SD-LEU agar plates; (**B**) Western blot of extracted proteins from cells used in panel (**A**) probed with IgG specific for Ded1 or PGK1. The positions of the molecular weight markers and proteins are as indicated. (**C**) Purified Ded1 was irradiated with UV light for the indicated times in minutes and separated on an SDS-PAGE. For experimental details, see the Material and Methods section.

**Figure 2 genes-14-01566-f002:**
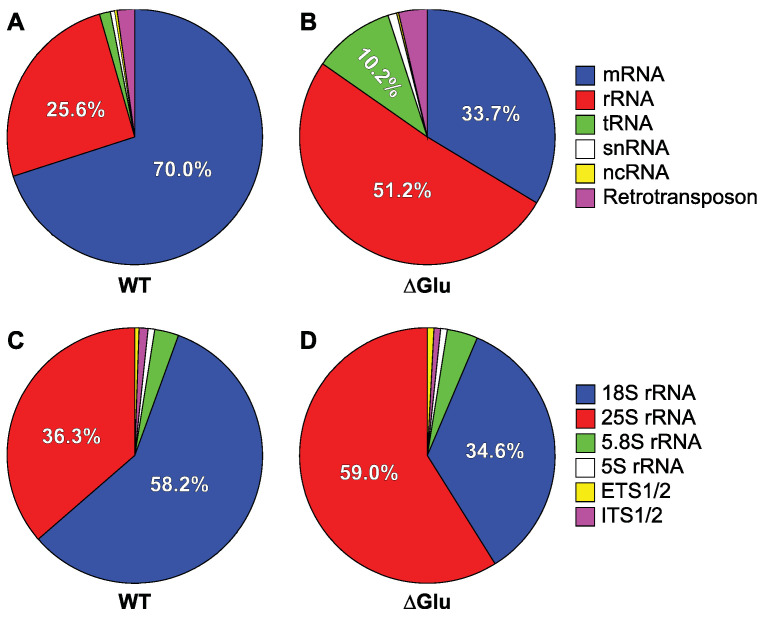
Distribution of recovered RNAs crosslinked to Ded1. The identities of the sectors are shown to the right for total RNAs (**A**,**B**) and ribosomal RNAs (**C**,**D**). (**A**) Ded1 crosslinked to RNAs in cells growing under standard (WT) conditions in the presence of glucose. (**B**) Ded1 crosslinked to RNAs in cells depleted for glucose (∆Glu). (**C**) Distribution of crosslinks on rRNAs in cells growing under standard conditions. (**D**) The distribution of rRNA crosslinks with glucose depletion.

**Figure 3 genes-14-01566-f003:**
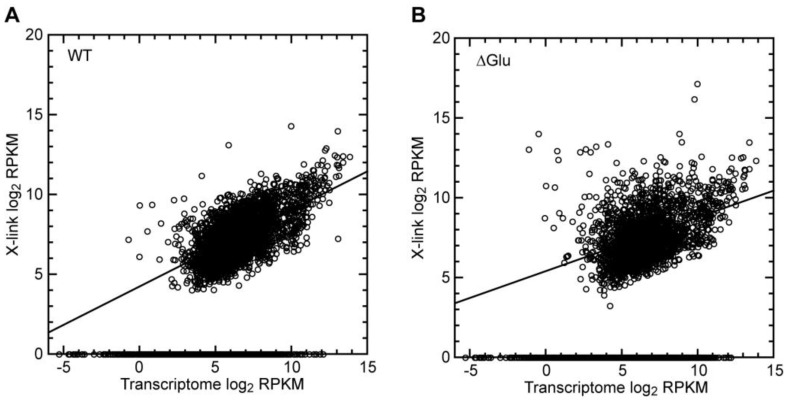
Crosslinks on Ded1 were partially correlated with mRNA abundance. (**A**) Distribution of the number of crosslinks relative to the total transcribed mRNA normalized for the size and quantity of the RNA (RPKM) for cells grown under standard conditions (WT), excluding pseudogenes, blocked reading frames, and retrotransposons. About 50% of the mRNAs showed at least one crosslinked fragment, and the slope of the curve fit was 0.480, with a correlation coefficient (R^2^) of 0.409. The transcriptome of the uncrosslinked mRNA was arbitrarily assigned a value of 0, and it is shown to reveal its distribution; it was not included in the linear regression. (**B**) The same as in panel **A** but with cells grown under glucose-depletion conditions (∆Glu). Only about 37% of the mRNAs showed at least a single crosslinked fragment. The curve fit was 0.336 with an R^2^ of 0.183.

**Figure 4 genes-14-01566-f004:**
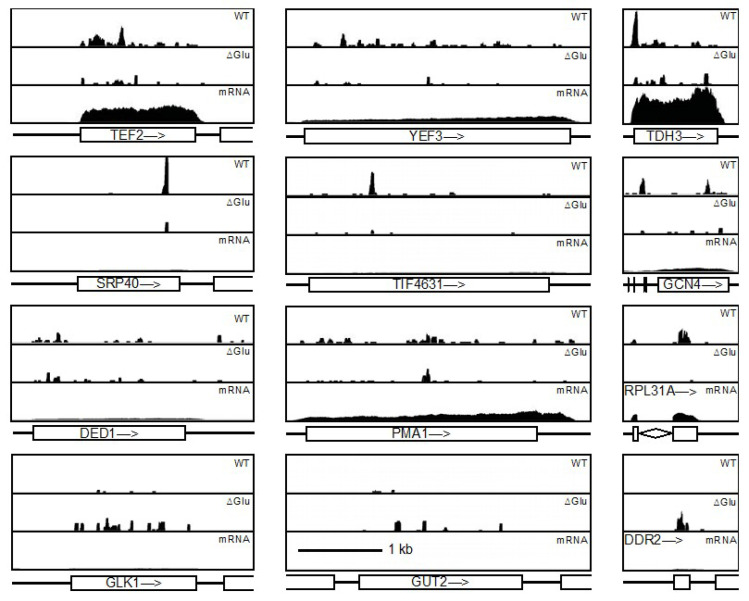
The distribution of Ded1 crosslinks on mRNAs was highly variable. The recovered crosslinked fragments of representative genes under standard (WT) and glucose-depletion (∆Glu) conditions. The corresponding transcriptomes (mRNA) are also shown. Panels are all drawn to the same scale, where the *Y*-axis is 30 for the crosslinks and 53,000 for the transcriptome. The scale bar for the *X*-axis is as shown in the *GUT2* panel. Data were analyzed using Integrated Genome Browser (IGB) and visualized with the Integrated Genome Viewer (IGV; [93]).

**Figure 6 genes-14-01566-f006:**
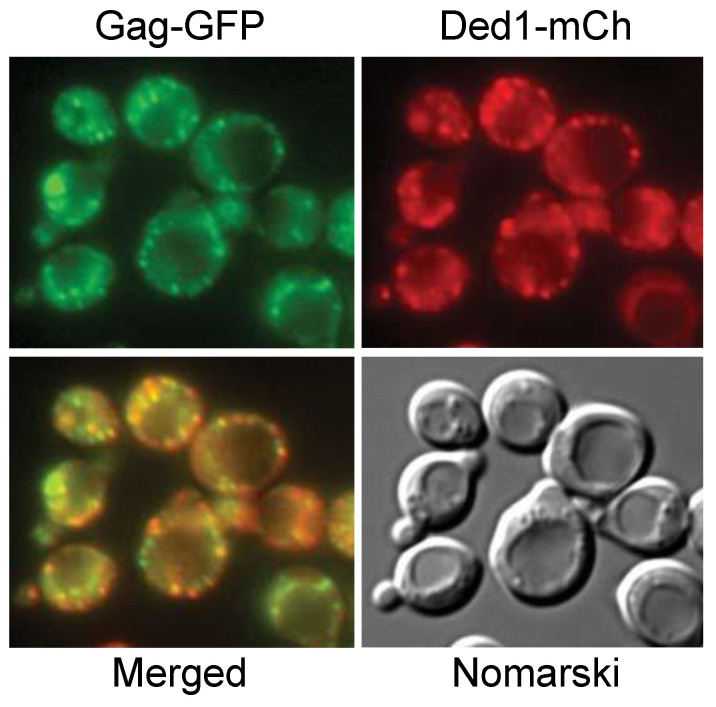
Ded1 colocalizes with Gag on the ER. G50 cells were incubated at 37 °C for 15 min to promote the formation of the foci containing Ded1. Both the wildtype and DQAD mutant of Ded1 provided similar results. In yeast, the ER envelops the nucleus (central cisternal ER) and then extends as a cortical halo inside the plasma membrane of the cell wall (PM-associated ER) through interconnected tubules (tubular ER; [111]).

**Figure 7 genes-14-01566-f007:**
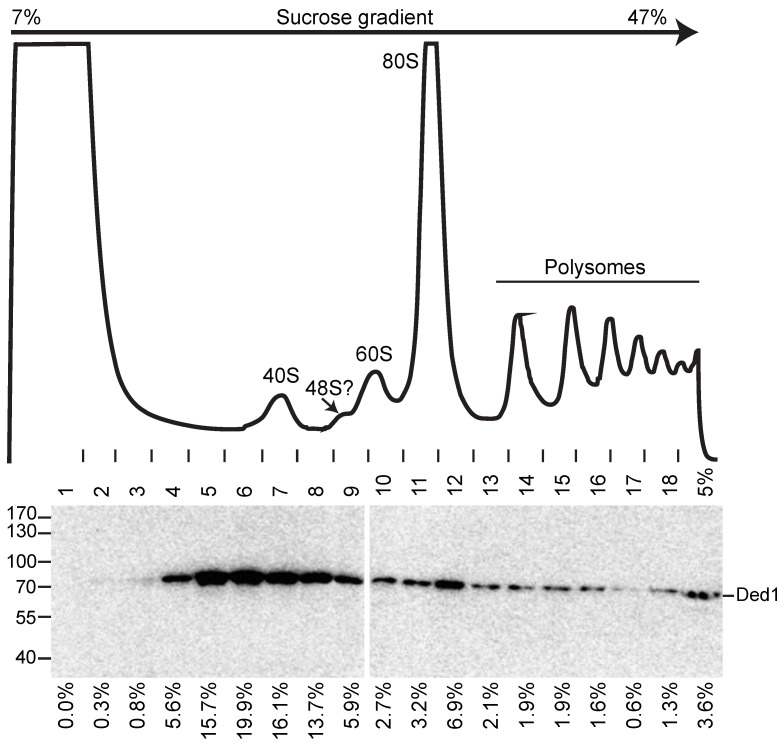
Distribution of Ded1 in a sucrose gradient. Cell extracts were incubated on ice with 5 mM AMP-PNP and GMP-PNP just prior to loading on the 7–47% sucrose gradient. Fractions were collected from the top and monitored at 254 nm. The very pronounced peak in fractions 1 and 2 is partially due to the free AMP-PNP and GMP-PNP. The Western blot of the SDS Laemmli gel shows the fractions containing Ded1. The last lane shows ~5% of the cell extract used for the gradient. The locations of the molecular weight markers and Ded1 are as shown. The percentages of the recovered material (lanes 1–18) are as shown below. Note that more than 70% of the inputted material (lane 5%) was recovered from the gradient (lane 5% divided by lanes 1–18). Further note that the vast majority of Ded1 sedimented above the position corresponding to the 48S PIC/IC.

**Table 1 genes-14-01566-t001:** Top 20 characterized crosslinked mRNAs upregulated with glucose depletion.

Gene	ORF	5′ UTR	3′ UTR	GO Term ^a^
*TAR1*	374	110	91	Cellular respiration (GO:0045333)
*DDR2*	185	34	244	Response to oxidative stress (GO:0006979)
*HKR1*	5408	31	142	Response to osmotic stress (GO:0006970)
*HSP26*	644	60	165	Protein folding (GO:0006457)
*SOL4*	767	35	96	Generation of precursor metabolites and energy (GO:0006091)
*TKL2*	2045	– ^2^	– ^2^	Generation of precursor metabolites and energy (GO:0006091)
*GLK1*	1502	31	180	Generation of precursor metabolites and energy (GO:0006091)
*GSY1*	2126	70	112	Generation of precursor metabolites and energy (GO:0006091)
*TEF1*	1376	33	123	Translational elongation (GO:0006414)
*YSC84*	1574	47	109	Cytoskeleton organization (GO:0007010)
*SPI1*	446	21	165	GPI-anchored cell wall protein
*GND2*	1478	– ^2^	114	Generation of precursor metabolites and energy (GO:0006091)
*AFG1*	1529	88	61	Response to oxidative stress (GO:0006979)
*HST3*	1343	43	149	Chromatin organization (GO:0006325)
*STF1*	260	53	164	Regulator of mitochondrial F1F0-ATP synthase
*ALD4*	1559	39	226	Generation of precursor metabolites and energy (GO:0006091)
*APE3*	1613	33	50	Vacuolar aminopeptidase Y
*SIL1*	1265	63	23	Protein targeting (GO:0006605)
*HBT1*	3140	– ^2^	– ^2^	Cell morphogenesis (GO:0000902)
*GAD1*	1757	39	105	Amino acid metabolic process (GO:0006520)

^a^ Multiple GO terms were obtained for some, while none were found for others (*SPI1*, *STF1*, *APE3*). ^2^ Value is not available.

**Table 2 genes-14-01566-t002:** Bottom 20 characterized crosslinked mRNAs downregulated with glucose depletion.

Gene	ORF	5′ UTR	3′ UTR	GO Term ^a^
*CCT7*	1652	166	130	Protein folding (GO:0006457)
*BFR1*	1412	– ^2^	200	Regulation of organelle organization (GO:0033043)
*RPL31A*	762	21	70	Cytoplasmic translation (GO:0002181)
*VMA2*	1553	46	148	Transmembrane transport (GO:0055085)
*GDH1*	1364	57	81	Amino acid metabolic process (GO:0006520)
*XRN1*	4586	333	124	RNA catabolic process (GO:0006401)
*RRP14*	1304	74	91	Ribosome assembly (GO:0042255)
*PBP1*	2168	215	101	Regulation of translation (GO:0006417)
*RPS14A*	720	22	78	rRNA processing (GO:0006364)
*TSR1*	2366	42	98	rRNA processing (GO:0006364)
*USO1*	5372	33	136	Vesicle organization (GO:0016050)
*NUM1*	8246	– ^2^	50	Cytoskeleton organization (GO:0007010)
*ENP2*	2123	48	35	rRNA processing (GO:0006364)
*MPP10*	1781	17	361	rRNA processing (GO:0006364)
*CAB4*	917	19	53	Small molecule metabolic process (GO:0055086)
*SUB2*	1340	156	111	RNA splicing (GO:0008380)
*SEC62*	824	12	44	Protein targeting (GO:0006605)
*TUB2*	1373	66	226	Cytoskeleton organization (GO:0007010)
*SEC61*	1442	206	144	Protein targeting (GO:0006605)
*MAL31*	1844	155	69	Maltose permease

^a^ Multiple GO terms were obtained for some, while none were found for others (*MAL31*). ^2^ Value is not available.

## Data Availability

Sequencing data can be retrieved using GEO accession number GSE228828. Go to https://www.ncbi.nlm.nih.gov/geo/query/acc.cgi?acc=GSE228828.

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
