# Peer review of "The DEAD-Box RNA Helicase Ded1 Is Associated with Translating Ribosomes"

_genes, 2023, doi:10.3390/genes14081566_

Round 1
Reviewer 1 Report
Major comments:
An exciting manuscript explores DEAD-box RNA helicase Ded1's role in translation. The authors employ qtPAR-CLIP methods to research the Ded1 interaction with open reading frames of many different mRNAs in yeast cells. Meanwhile, under glucose-depletion conditions, the crosslinking pattern shifts to mRNAs encoding 23 metabolic and stress-related proteins, which reflects the altered translation. This manuscript provided a novel insight into the function of Ded1. The manuscript has several mistakes, such as lines 145, 992, and 1001. The authors should check it carefully. Thanks.
Minor comments:
Line 992: Please remove the sentence segment “Please add”
Line 1001: Please remove the word “I”.
Sufficient
Author Response
We thank the reviewer for their encouraging remarks.
"The manuscript has several mistakes, such as lines 145, 992, and 1001."
We made the appropriate corrections
Line 992: Please remove the sentence segment “Please add”
We made the appropriate correction
Line 1001: Please remove the word “I”.
We made the appropriate correction
Reviewer 2 Report
This paper presents findings on Ded1, a yeast DEAD-box RNA helicase, and its interactions with different mRNA targets using a modified PAR-CLIP technique. The study reveals that Ded1 forms extensive interactions on select mRNAs, particularly those encoding ribosomal proteins and translation factors, suggesting its involvement in translation regulation. Overall, this manuscript provides valuable insights into the interactions and potential role of Ded1 in translation regulation. The findings are significant and need to addresse the following minor revisions:
Figure 1: The figure requires clarification. The legend mentions two "A" letters, which is confusing. It would be helpful to mention the type of cell culture, incubation days, and image capture details. The numbers on the left and right sides of the figure need to be clearly labeled for specific information. Additionally, labeling the graphs as UV-A and UV-C instead of 254nm and 365nm would enhance clarity.
Line 366: The statement "For experimental details, see the Material and Methods section" is unnecessary in the legend of Figure 1.
Figure 3: The legend for Figure 3, specifically part B, should be rewritten to provide clearer information.
Figure 7: Please label the protein names in the western blot bands. It would be beneficial to include quantification of the western blot bands for greater precision and convincing evidence.
Remove the phrase "reviewed by" from some text citations.
Line 460-461: The statement "In contrast, we obtained fewer crosslinked fragments of SRG1 under the glucose-depletion conditions" should be cited with the corresponding figure.
Addressing these minor revisions will enhance the clarity and overall quality of the manuscript.
The English language can be improved.
Author Response
We thank the reviewer for their corrections and positive remarks.
Figure 1: The figure requires clarification. The legend mentions two "A" letters, which is confusing. It would be helpful to mention the type of cell culture, incubation days, and image capture details. The numbers on the left and right sides of the figure need to be clearly labeled for specific information. Additionally, labeling the graphs as UV-A and UV-C instead of 254nm and 365nm would enhance clarity.
We added "panel" to clarify that the cells used in panel B were the same as those used in panel A. We added the labeling to the graphs and used UV-A and UV-C instead of 254nm and 365nm.
Line 366: The statement "For experimental details, see the Material and Methods section" is unnecessary in the legend of Figure 1.
We deleted this line.
Figure 3: The legend for Figure 3, specifically part B, should be rewritten to provide clearer information.
Panel B is the same as panel A except for cells grown under glucose-depletion conditions. We added some clarification, but we are not sure what further elaboration the reviewer is seeking.
Figure 7: Please label the protein names in the western blot bands. It would be beneficial to include quantification of the western blot bands for greater precision and convincing evidence.
We have added this information to the figure.
Remove the phrase "reviewed by" from some text citations.
This is often an editorial choice. We use this phrase to distinguish between published results and résumés thereof. We think that this is useful to the reader.
Line 460-461: The statement "In contrast, we obtained fewer crosslinked fragments of SRG1 under the glucose-depletion conditions" should be cited with the corresponding figure.
We have added this information to the text.
Reviewer 3 Report
Yeter-Alat et al. have developed a streamlined variant of the PAR-CLIP method that they term quicktime PAR-CLIP (qtPAR-CLIP) that involves irradiating cells that had incorporated 4-thiouridine into RNA with UV-A light (365 nm peak). The method avoids the use of harsh UV-C (260 nm) that kills cells and induces degradation of two tested proteins (Ded1p and PGK2). As well allowing a faster turnaround, this method thus also has the advantage of being more likely to capture RNP complexes in their intact state.
The authors used qtPAR-CLIP to identify RNAs that interact with the yeast Ded1p protein and to map the sites on them to which this protein binds. Consistent with its prior identification as a translation factor, Ded1p was primarily bound to mRNA and 18S rRNA, and this association was reduced two-fold in conditions (glucose deprivation) that rapidly downregulate translation initiation. This change was associated with a partial shift in Ded1p association from ribosomal 40S subunits to 80S ribosomes (primarily due to reduced level of cross-linking to 18S rRNA), although the mechanistic basis for this shift was not determined. Indeed, the location of cross-linked nucleotides suggested binding to more than one site. Ded1p was associated with only a subset of mRNAs, and this number decreased in glucose-depletion conditions, but in both conditions, binding of Ded1p was restricted to actively translated mRNAs. Cross-linking sites were concentrated at specific purine-rich motifs in the coding regions of specific mRNAs, which they authors suggest may reflect a function of Ded1p in pausing translocating ribosomes or stabilizing them after arrest. It is thus not obvious why the authors suggest that Ded1p is an elongation factor.
In conclusion, the qtPAR-CLIP method is technically a useful advance, but the data presented here obtained using it are primarily descriptive, so that some conclusions are overly definitive and should be toned down. Nevertheless, the findings are provocative and will likely lead to a re-assessment of the function of Ded1p in translation.
SPECIFIC COMMENTS.
1. The title is too definitive: the authors do not prove that Ded1p is a translation elongation factor, and they should not make such a claim without conclusive experimental support.
2. The PAR-CLIP approach involves immunoprecipitation, and Yeter-Alat et al. frequently refer to pre-immune IgG and IgG specific for Ded1 or PGK1 (e.g. lines 200-201, 204-205, 208, 353-354, 364, 387), but they do not specify the source of these critical reagents in Methods. This must be corrected.
3. The authors should use the accepted nomenclature when referring to the protein Ded1p. The convention (e.g. SGD (Saccharomyces Genome Database) http://genome-www.stanford.edu/Saccharomyces/) is that the names of genes in S. cerevisiae comprise three letters and an arabic number (e.g. DED1) whereas the corresponding protein is “referred to by the relevant gene symbol, non-italic, initial letter uppercase and with the suffix ‘p’” .. in this paper, Ded1p.
4. The authors describe the association of Ded1p with specific sites on a small subset of mRNAs (lines 590-609 and Figure 4). It would be more informative if they presented a global analysis of Ded1p binding sites, specifically detailing the density of binding to 5’UTR, principal ORF, uORFs and 3’UTR after normalization for length. Is the preferential association with coding sequences in specific mRNAs a general characteristic?
5. Has part of Figure 4 on the right-hand side been omitted/truncated?
6. The authors claim (lines 725-726) that tRNAs activate the RNA-dependent ATPase activity of Ded1p. Data supporting this claim are not shown. Have they been published elsewhere? This should be clarified.
7. The authors’ research concerns yeast ribosomes, and their statement that ribosomes have helicase activity (lines 779-780) will be understood to refer to eukaryotic ribosomes. However, this is misleading, because the helicase activity of ribosomes has only been characterized as a property of bacterial ribosomes, so this section should be clarified.
8. The authors repeatedly refer to 40S ribosomes, 48S ribosomes and 60S ribosomes, but this nomenclature is incorrect and misleading. 40S and 60S refer to subunits of the 80S ribosome, whereas 48S refers to initiation complexes containing mRNA and a 40S subunit. The text should be corrected throughout.
9. Typographical and other minor errors
a) Line 122. “BY4742” is not strictly a wild type strain, but rather, is a deletion strain derived from S228C.
b Line 201” “immunoglubuline” should be “immunoglobuin”
c)Line 248. Manufacture’s should be manufacturer’s
d) Line 426. “glucose-deletion” should be “glucose-depletion”
e) Line 502. What are “potential mRNAs”? Do the authors mean half of all mRNAs?
Specific comment #9.
b Line 201” “immunoglubuline” should be “immunoglobuin”
c)Line 248. Manufacture’s should be manufacturer’s
d) Line 426. “glucose-deletion” should be “glucose-depletion”
e) Line 502. What are “potential mRNAs”? Do the authors mean half of all mRNAs?
Author Response
We thank the reviewer for their thorough reading of the text and for revealing the typos and ambiguities that we missed.
- The title is too definitive: the authors do not prove that Ded1p is a translation elongation factor, and they should not make such a claim without conclusive experimental support.
The reviewer is correct that we have not shown direct evidence for the role of Ded1 in translation elongation, but this is also true for its role in translation initiation. Factors involved in translation elongation can regulate the process in multiple ways, and they can both increase and decrease the elongation rates. In particular, we previously found that the ATPase activity of Ded1 is negatively regulated by the signal recognition particle, which are involved in pausing the ribosomes and permitting them to associate with the endoplasmic reticulum. A biorxiv preprint of this work is available (https://doi.org/10.1101/2020.11.08.373522). Thus, our results are consistent with Ded1 playing this role. Nevertheless, we have modified the title to better reflect the ambiguities.
- The PAR-CLIP approach involves immunoprecipitation, and Yeter-Alat et al. frequently refer to pre-immune IgG and IgG specific for Ded1 or PGK1 (e.g. lines 200-201, 204-205, 208, 353-354, 364, 387), but they do not specify the source of these critical reagents in Methods. This must be corrected.
The reviewer is quite right that we neglected to clarify this. The information was present in the Supplementary Data and in the biorxiv preprint, but the information was shifted during editing. We have added the requested information in the Materials and Methods section.
- The authors should use the accepted nomenclature when referring to the protein Ded1p. The convention (e.g. SGD (Saccharomyces Genome Database) http://genome-www.stanford.edu/Saccharomyces/) is that the names of genes in S. cerevisiae comprise three letters and an arabic number (e.g. DED1) whereas the corresponding protein is “referred to by the relevant gene symbol, non-italic, initial letter uppercase and with the suffix ‘p’” .. in this paper, Ded1p.
The reviewer is correct that yeast geneticists have a unique genetic nomenclature (as is often the case with other organisms). However, we work with, and make comparisons to, a number of different organisms, which have different conventions. For example, it is yeast Ded1p and mammalian DDX3. Therefore, we have used more universal genetic terms (basically as delineated by HUGO Gene Nomenclature) throughout the text. Italic uppercase for wildtype genes, italic lowercase for mutant genes and non-italic case for the proteins. We have endeavored to be a consistent in this in all our publications. We feel this makes our work more accessible to a wider readership.
- The authors describe the association of Ded1p with specific sites on a small subset of mRNAs (lines 590-609 and Figure 4). It would be more informative if they presented a global analysis of Ded1p binding sites, specifically detailing the density of binding to 5’UTR, principal ORF, uORFs and 3’UTR after normalization for length. Is the preferential association with coding sequences in specific mRNAs a general characteristic?
We tried to present representative data in Figure 4, which shows a wide range of distributions. In contrast to previous publications using CRAC/CLIP, we recovered very few crosslinks in the 5' UTRs, as well as very few in the 3' UTRs. From Figure 4 it is clear that Ded1 associates with the different mRNAs in widely different fashions that range from a broad distribution (e.g., YEF3, PMA1) to very narrow (e.g., SRP40). Thus, a meaningful global analysis of the Ded1 binding sites is difficult (and would be misleading). Nevertheless, Figure 3 shows that there is a weak correlation between the density of binding sites (length and quantity) of RNA and the recovered crosslinked fragments. We note that all of these data are freely available on the GEO database site.
- Has part of Figure 4 on the right-hand side been omitted/truncated?
The PDF version of the manuscript does indeed show part of Figure 4 cut off. It is correct in the original Word version. We assume Genes will make the appropriate corrections.
- The authors claim (lines 725-726) that tRNAs activate the RNA-dependent ATPase activity of Ded1p. Data supporting this claim are not shown. Have they been published elsewhere? This should be clarified.
The reviewer is correct in that this information is in another paper submitted (hence unpublished). We include a modified image from this paper for the reviewer's benefit (not to be published here). The ATPase activity of Ded1 is stimulated by a number of different RNAs to varying extents; however, extensive analyses thereof is beyond the scope of this manuscript. The point here was only that Ded1 binds tRNAs as well as other RNAs.

- The authors’ research concerns yeast ribosomes, and their statement that ribosomes have helicase activity (lines 779-780) will be understood to refer to eukaryotic ribosomes. However, this is misleading, because the helicase activity of ribosomes has only been characterized as a property of bacterial ribosomes, so this section should be clarified.
The reviewer is correct in that this work was done with prokaryotic ribosomes, although it is considered applicable to the eukaryotic ribosomes as well. We added a clarifying statement.
- The authors repeatedly refer to 40S ribosomes, 48S ribosomes and 60S ribosomes, but this nomenclature is incorrect and misleading. 40S and 60S refer to subunits of the 80S ribosome, whereas 48S refers to initiation complexes containing mRNA and a 40S subunit. The text should be corrected throughout.
We were very careful in our usage of this terminology. The dissociated 40S ribosomes are associated with various factors to form the 42S/43S ribosomes that subsequently associate with the eIF4F-mRNA complex to form the 48S PIC. This complex scans the 5'UTR to the AUG start codon whereupon it undergoes changes to form the 48S IC, which subsequently undergoes conformational changes to associate with the 60S ribosomes and form the 80S ribosomes that are competent for translation. We partially detailed this information in the Introduction. However, as we mentioned in the text, we are unable to distinguish crosslinks on the 48S PIC/IC and the 40S ribosomes associated with the 60S ribosomes. There were no crosslinked positions unique to the 48S PIC/IC. Nevertheless, the 40S core structure is present in all of these complexes. We have clarified this within the text where necessary.
- Typographical and other minor errors
- a) Line 122. “BY4742” is not strictly a wild type strain, but rather, is a deletion strain derived from S228C.
b Line 201” “immunoglubuline” should be “immunoglobuin”
c)Line 248. Manufacture’s should be manufacturer’s
- d) Line 426. “glucose-deletion” should be “glucose-depletion”
- e) Line 502. What are “potential mRNAs”? Do the authors mean half of all mRNAs?
The reviewer is correct that BY4742 has a deletions and mutations that facilitate genetic selection, but it is considered virtually identical to the parent strain (Brachmann et al, 1998, Yeast 14, 115–132). However, there is some question about whether any laboratory strain of S. cerevisiae can truly be considered wild type. We leave this to the reader to decide.
We have corrected the indicated typos.
Potential refers to the transcribed genes. However, we deleted "potential" since it is redundant.
Round 2
Reviewer 3 Report
The authors have responded satisfactorily to most comments and suggestions, but they should still revise their manuscript in respond to the following previous comments.
Comment #5
Figure 4 still appears to have been truncated. The authors must ensure that the production team corrects the manuscript so that the complete image is shown.
Comment #6
The authors state that tRNAs activate the RNA-dependent ATPase activity of Ded1p (lines 733-734 of the revised manuscript). However, these data have not previously been published nor are they presented in the present manuscript.
The instructions to authors for “Genes” states:
"Unpublished data" intended for publication in a manuscript that is either planned, "in preparation" or "submitted" but not yet accepted, should be cited in the text and a reference should be added in the References section.
The authors must revise their manuscript appropriately.
Comment #8.
Despite their protestation “We were very careful in our usage of this terminology”, the authors have not revised the manuscript appropriately in response to this comment and still refer e.g. to 40S ribosomes (lines 763, 813, 845 etc) and 60S ribosomes (e.g. lines 882-883, 902 etc). Such entities do not exist.
The ribosome has a Svedberg coefficient of 80 (thus, 80S), but 40S and 60S refer to subunits of the ribosome, not to ribosomes themselves. They must refer to 40S ribosomal subunits and 60S ribosomal subunits.